# Mechanism of bisphosphonate-related osteonecrosis of the jaw (BRONJ) revealed by targeted removal of legacy bisphosphonate from jawbone using competing inert hydroxymethylene diphosphonate

**Hiroko Okawa[1,2†], Takeru Kondo[1,2†], Akishige Hokugo[1,3]\*, Philip Cherian[4], Jesus J Campagna[5], Nicholas A Lentini[6], Eric C Sung[1], Samantha Chiang[7], Yi-Ling Lin[8], Frank H Ebetino[4], Varghese John[5], Shuting Sun[1,4]\*, Charles E McKenna[6]\*, Ichiro Nishimura[1,7]\***

[1]Weintraub Center for Reconstructive Biotechnology, Division of Regenerative & Reconstructive Sciences, University of California, Los Angeles School of Dentistry, Los Angeles, United States; [2]Division of Molecular & Regenerative Prosthodontics, Tohoku University Graduate School of Dentistry, Sendai, Japan; [3]Regenerative Bioengineering and Repair Laboratory, Division of Plastic and Reconstructive Surgery, Department of Surgery, David Geffen School of Medicine at University of California, Los Angeles, Los Angeles, United States; [4]BioVinc, LLC, Pasadena, United States; [5]Department of Neurology, David Geffen School of Medicine at University of California, Los Angeles, Los Angeles, United States; [6]Department of Chemistry, University of Southern California, Los Angeles, United States; [7]Division of Oral & Systemic Health Sciences, University of California, Los Angeles School of Dentistry, Los Angeles, United States; [8]Section of Oral & Maxillofacial Pathology, University of California, Los Angeles School of Dentistry, Los Angeles, United States

**\*For correspondence:**
ahokugo@mednet.ucla.edu (AH);
shuting.sun@biovinc.com (SS);
mckenna@usc.edu (CEMcK);
inishimura@dentistry.ucla.edu (IN)

†These authors contributed equally to this work

**Abstract** Bisphosphonate-related osteonecrosis of the jaw (BRONJ) presents as a morbid jawbone lesion in patients exposed to a nitrogen-containing bisphosphonate (N-BP). Although it is rare, BRONJ has caused apprehension among patients and healthcare providers and decreased acceptance of this antiresorptive drug class to treat osteoporosis and metastatic osteolysis. We report here a novel method to elucidate the pathological mechanism of BRONJ by the selective removal of legacy N-BP from the jawbone using an intra-oral application of hydroxymethylene diphosphonate (HMDP) formulated in liposome-based deformable nanoscale vesicles (DNV). After maxillary tooth extraction, zoledronate-treated mice developed delayed gingival wound closure, delayed tooth extraction socket healing and increased jawbone osteonecrosis consistent with human BRONJ lesions. Single cell RNA sequencing of mouse gingival cells revealed oral barrier immune dysregulation and unresolved proinflammatory reaction. HMDP-DNV topical applications to nascent mouse BRONJ lesions resulted in accelerated gingival wound closure and bone socket healing as well as attenuation of osteonecrosis development. The gingival single cell RNA sequencing demonstrated resolution of chronic inflammation by increased anti-inflammatory signature gene expression of lymphocytes and myeloid-derived suppressor cells. This study suggests that BRONJ pathology is related to N-BP levels in jawbones and demonstrates the potential of HMDP-DNV as an effective BRONJ therapy.

## Editor's evaluation

The manuscript shows that bisphosphonate-related osteonecrosis of the jaw, a rare complication of osteoporosis treatment and bone marrow cancers, was prevented/alleviated in mice using a novel treatment which works by reversing the associated oral inflammation. The work in this manuscript is valuable and will be of significant interest to investigators in the bone and dental fields who conduct pre-clinical research.

## Introduction

Nitrogen-containing bisphosphonates (N-BPs) are prototypical antiresorptive agents (*Ebetino et al., 2011*; *Cremers et al., 2019*; *McKenna et al., 2020*) initially marketed to prevent bone fractures and to treat osteopenia or osteoporosis (*Fink et al., 2019*; *Black and Rosen, 2016*). N-BPs have been widely prescribed for postmenopausal women who have a bone mineral density T score of −2.5 or less, a history of spine or hip fracture, or a Fracture Risk Assessment Tool score (*Kanis et al., 2007*) indicating increased fracture risk. N-BP treatment with an increased dose and frequency is also given to patients with multiple myeloma (*Mhaskar et al., 2017*) or metastatic cancers to bone (*O'Carrigan et al., 2017*; *Goldvaser and Amir, 2019*) to address tumor-induced osteolysis, hypercalcemia, and bone pain. N-BPs are well tolerated, exhibit few side-effects, and have established clinical benefits (*Gordon, 2005*).

In the 2000's, cases of osteonecrosis in the jawbone (ONJ) emerged among a minority of patients with a history of N-BP treatment, usually occurring at a dental infection site (*Khan et al., 2015*) or after dentoalveolar surgery such as a tooth extraction (*Marx, 2003*; *Ruggiero, 2009*). Originally designated as 'bisphosphonate-related ONJ' (BRONJ), this syndrome has also proven to be associated with non-bisphosphonate antiresorptive agents such as denosumab, a humanized anti-RANKL monoclonal antibody and anti-angiogenesis drug. As a result, the American Association for Oral & Maxillofacial Surgeons (AAOMS) has proposed the term 'Medication-related ONJ' (MRONJ) reflecting the association of ONJ with a multiplicity of antiresorptive agents (*Ruggiero et al., 2014*). Incidents of MRONJ reported to the United States Food and Drug Administration (FDA)'s Adverse Event Reporting System (FAERS) peaked from the first quarter of 2010 to the first quarter of 2014 with approximately 30,000 cases during this period, among which BRONJ represented the major fraction (*Zhang et al., 2016*). The FAERS database may underreport incidents because of variations in provider's perceptions of how severe an event needs to be to warrant submission of the event report (*Alatawi and Hansen, 2017*). We present an alternative MRONJ estimation of approximately 23,511 new cases per year in the US (*Table 1*). Considering the long cure period, the cumulative MRONJ patient number may exceed this annual estimation; however, it is highly conceivable that MRONJ is a rare disease defined by the FDA's orphan disease (*Kempf et al., 2018*).

**Table 1.** Estimated MRONJ case numbers in the US based on MRONJ incidence for the major underlying diseases[*].

| Underlying diseases | New cases in the US (Year) | MRONJ incidence | Estimated MRONJ cases (Year)[*] |
|---|---|---|---|
| Multiple myeloma | 34,920[†] | 5.16% (*Rugani et al., 2016*) | 1,802 |
| Breast cancer | 330,840[‡] | 2.09% (*Rugani et al., 2016*) | 6,915 |
| Prostate cancer | 248,530[†] | 3.80% (*Rugani et al., 2016*) | 9,444 |
| Osteoporosis/Low bone mass | 53,500,000[§] | 0.01% (*Khan et al., 2017*) | 5,350 |
| Estimated annual incidents of MRONJ | | | 23,511 |

MRONJ: Medication-related osteonecrosis in the jawbone.

[*]Based on an assumption that all these patients were treated by antiresorptive medications.

[†]American Cancer Society. Cancer Facts & Figures 2021. Atlanta, Ga: American Cancer Society; 2021.

[‡]Invasive breast cancer and ductal carcinoma: American Cancer Society. How Common Is Breast Cancer? Jan. 2020. Available at: https://www.cancer.org/cancer/breast-cancer/about/how-common-is-breast-cancer.html.

[§]Center for Disease Control and Prevention at: https://www.cdc.gov/nchs/products/databriefs/db405htm.

Typical disease symptoms include gingival dehiscence and exposure of necrotic jawbone (*Ruggiero et al., 2014*; *Reid and Cornish, 2011*; *Otto et al., 2018*), which often becomes infected, resulting in pain, erythema, and purulent drainage (*Ruggiero, 2015*). The extent of a BRONJ lesion has been measured for the absolute or relative open wound area and the time taken for mucosa to completely cover necrotic tissue and exposed bone ('cure period') (*Beth-Tasdogan et al., 2017*). A multicenter case registry study reported either resolution (35%) or improvement (10%) in 207 evaluable BRONJ patients within the study period of 2 years with conservative treatments such as irrigation and antibiotic medications (*Schiodt et al., 2018*). However, 37% of patients did not respond to the conventional treatment and exhibited either progression or stable conditions. Although the reported case incident is small, long-lasting severe clinical symptoms of MRONJ in some patients have created apprehension among patients and healthcare providers (*Kelly et al., 2019*; *Jha et al., 2015*).

The National Institutes of Health (NIH) workshop on 'Pathway to Prevention (P2P) for Osteoporotic Fracture' in 2018 highlighted the acute need of research to mitigate serious adverse events such as MRONJ and atypical femur fracture to prevent the increasing threat of osteoporotic fractures (*Siu et al., 2019*). The P2P workshop particularly highlighted the markedly decreased acceptance of antiresorptive medications by osteoporosis patients, despite the significant benefit of these drugs for this indication (*van de Laarschot et al., 2020*). The recent decline in N-BP prescription, reflecting diminished patient acceptance of these drugs, has been linked to a statistical rise in bone-related complications (*Williams et al., 2015*). Due to the long-lasting half-life of N-BPs once chemisorbed to bone, the discontinuation of N-BPs and switching to other antiresorptive medications may still pose a risk of developing BRONJ (*Anagnostis et al., 2017*). It is therefore urgent to reduce the BRONJ risk associated with these otherwise effective medications for pathological osteolysis.

The mechanism of BRONJ is not yet fully understood (*Chang et al., 2018*). BRONJ exclusively occurs in the jawbone, which uniquely associated with the gingival oral barrier tissue with one of the most active barrier immunities (*Moutsopoulos and Konkel, 2018*). The presence of N-BP on the jawbone is shared by all BRONJ patients (*Lo Faro et al., 2019*) and it is conceivable that jawbone N-BP may uniquely interact with oral barrier immunity. The major challenge has been the difficulty in separating the systemic and oral effects of N-BP therapy, which has severely hindered the elucidation of the mechanisms of BRONJ.

N-BP chemisorption to bone results from its high affinity to hydroxyapatite (HAp), the mineral component of bone (*Nancollas et al., 2006*). Recent investigations in our laboratories revealed that pre-adsorbed N-BPs can be displaced from HAp by exposure to a second N-BP/BP in aqueous buffer both in vitro and in vivo (*Hokugo et al., 2013*; *Hokugo et al., 2019*). This led us to envision a new experimental model based on the selective removal of pharmacologically active legacy N-BP locally from jawbone by intra-oral application of a second BP with low pharmacological potency (lpBP). Here, we describe a liposome-based deformable nanoscale vesicle (DNV) formulation for topical trans-oral mucosa delivery of an lpBP to the underlying jawbone, and report its effect in attenuating N-BP-induced oral lesions using a murine BRONJ model and the elucidation of the mechanism involved in this pathology.

## Results

### Hydroxymethylene diphosphonate (HMDP, also known as hydroxymethylene bisphosphonate) was selected as the lpBP

N-BPs such as zoledronate (ZOL) act on osteoclasts by inhibiting farnesyl diphosphate synthase (FPPS), thus interfering with the mevalonate pathway and protein prenylation (*Russell, 2007*). The nitrogen-containing side chain of N-BPs, an essential pharmacophore in all potent N-BP antiresorptive drugs, is absent in HMDP which, however, retains an α-OH group together with two phosphonate groups, conferring high affinity to HAp (*Figure 1A*). Thus, we postulated that HMDP in solution might be able to decrease the amount of legacy N-BP at the jaw bone surface by a competitive displacement mechanism as we reported (*Hokugo et al., 2013*; *Hokugo et al., 2019*). ZOL IV injection to mice increased femur trabecular bone volume over the saline vehicle solution injection control, whereas HMDP IV injection with the same dose did not alter the bone architecture (*Figure 1B and C*). The lack of pharmacological effect of HMDP in vivo was consistent with our previous study using a standard in vitro prenylation assay (*Hokugo et al., 2019*) and confirmed HMDP to be an lpBP.

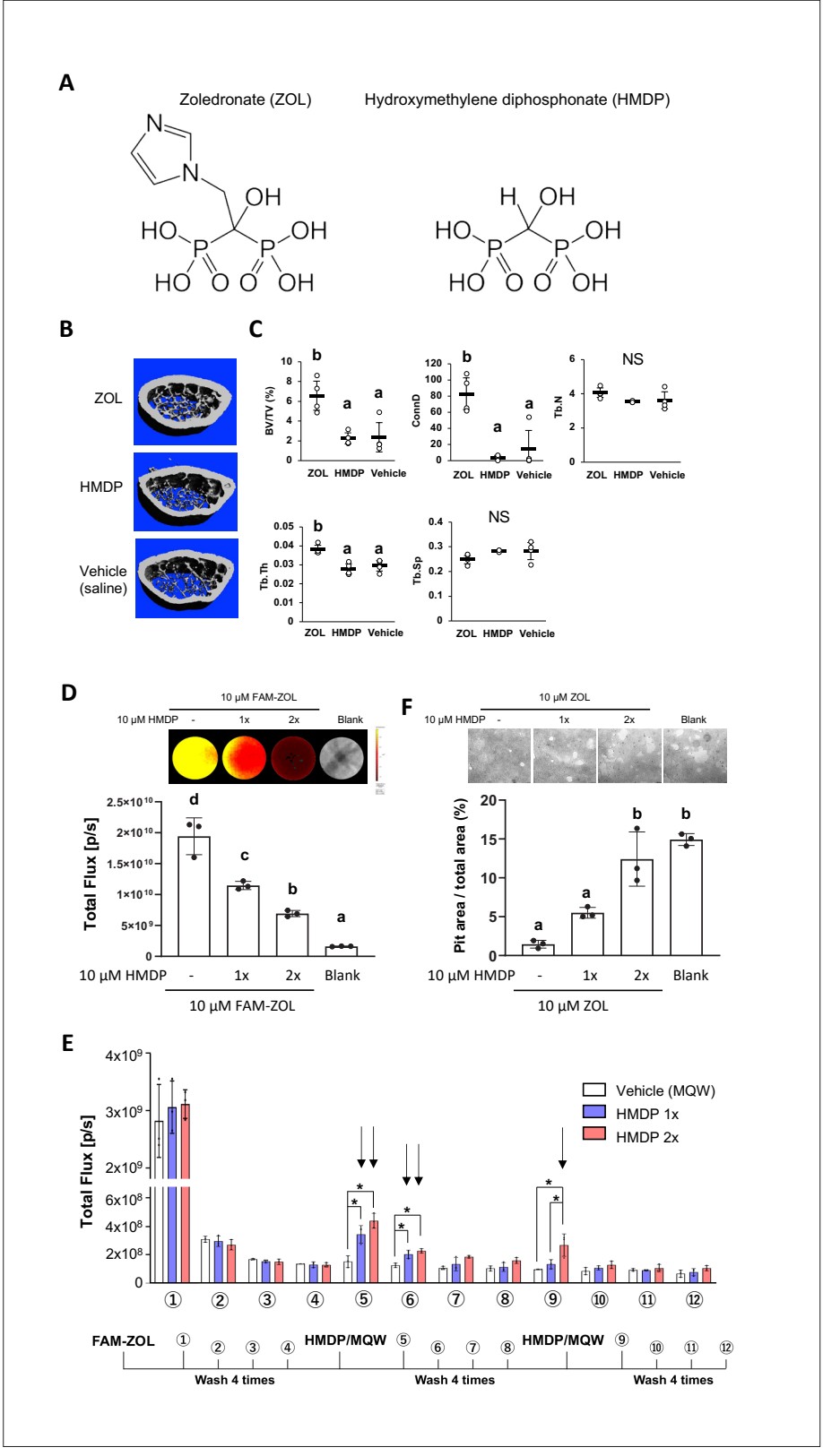

**Figure 1.** Competitive equilibrium-based dissociation of N-BP by low potency BP (lpBP). (**A**) Chemical structures (shown as the tetraacids) of N-BP, zoledronate (ZOL) and hydroxymethylene diphosphonate (HMDP). (**B**) ZOL but not HMDP affects mouse femur trabecular bone architecture. Mice received a bolus intravenous injection of 100 µl ZOL (40 nmol), HMDP (40 nmol) or vehicle saline (0.9% NaCl) solutions. Femurs (n=4 per group) were

*Figure 1 continued on next page*

*Figure 1 continued*

harvested 3 weeks after the IV injection and subjected to Micro-CT imaging. (**C**) HMDP did not affect the femur trabecular bone micro-architecture, whereas ZOL increased bone volume over total volume (BV/TV), connectivity density (ConnD) and trabecular thickness (Tb.Th). Trabecular number (Tb.N) and trabecular separation (Tb.Sp) measurements were not affected by ZOL. (*Figure 1—source data 1*) (**D**) In vitro demonstration of competitive displacement of legacy N-BP. Synthetic apatite-coated wells preincubated with 10 µM of 5-carboxyfluorescein-conjugated zoledronate (FAM-ZOL) were washed with MilliQ-treated pure water (MQW), then treated with 10 µM HMDP once (1×) or twice (2×) (n=3 per group). The FAM fluorescent signal measurement indicated significant reduction of the FAM-ZOL amount on the synthetic apatite. (*Figure 1—source data 2*) (**E**) Fluorescent signal measurement of the wash solutions from the in vitro experiment in (**D**) demonstrated removal of FAM-ZOL by the HMDP treatments (arrows). (*Figure 1—source data 3*) (**F**) In vitro osteoclastic pit formation assay. Synthetic apatite coated wells were preincubated with 10 µM ZOL followed by 10 µM HMDP treatment 1× or 2× (n=3 per group). RAW264.7 cells ($2.5×10^4$ cells per well) were then inoculated to each well in culture medium supplemented by mouse recombinant receptor activator of nuclear kappa-B ligand (RANKL). The areas of resorption pits generated by osteoclasts derived from RAW 264.7 cells were measured after 6 days of incubation and cell removal. Twice repeated HMDP treatments restored normal in vitro resorption pit formation. (*Figure 1—source data 4*) In (**D**) and (**F**), the graphs show the mean and SD (n=3 per group), and the Turkey test was used to analyze multiple samples. The statistical significance was determined to be at p<0.05. In (**E**), the graphs show the mean and SD (n=3 per group), and the Turkey test was used to analyze multiple samples within each time point. The statistical significance was determined to be at p<0.05. Different letters (e.g., a, b) are used to show statistically significant differences between multiple groups.

The online version of this article includes the following source data for figure 1:

**Source data 1.** Source data of *Figure 1C*.

**Source data 2.** Source data of *Figure 1D*.

**Source data 3.** Source data of *Figure 1E*.

**Source data 4.** Source data of *Figure 1F*.

Next, we tested the ability of HMDP to displace the pre-adsorbed N-BP from bone mineral surface. Synthetic apatite (carbonate apatite)-coated culture wells were pretreated with fluorescently tagged ZOL (FAM-ZOL: 10 µM). After thorough washing to remove non-chemisorbed FAM-ZOL, the wells were challenged by application of HMDP (10 µM in Milli-Q treated pure water: MQW). FAM-ZOL on the synthetic apatite was reduced by half (*Figure 1D*), and the wash solution contained a signal of dissociated FAM-ZOL (*Figure 1E*). After the second HMDP application, the amount of FAM-ZOL on the synthetic apatite was further reduced (*Figure 1D*). This experiment confirmed that a chemisorbed N-BP could be displaced by repeated applications of HMDP (*Figure 1D and E*).

We further examined the effect of competitive removal of ZOL on osteoclastic bone resorption in vitro. ZOL chemisorbed on synthetic apatite inhibited osteoclastic bone resorption in vitro as expected, which was measured by the decreased resorption pit area created by RAW 264.7-derived osteoclasts (*Figure 1F*). A single application of HMDP increased the pit area, albeit at no statistical significance. When HMDP was applied twice, the resorption pit size significantly increased and reached the level of ZOL-untreated blank wells (*Figure 1F*). Although the FAM-ZOL experiment suggested that ZOL would not completely be removed after two applications of HMDP, osteoclastic activity was restored to a nearly normal level as ZOL-untreated control group (*Figure 1F*). We postulate that there may be a threshold bioavailable concentration of N-BP causing osteoclast abnormality. Therefore, our goal is not to be complete removal of legacy N-BP from the jawbone, but rather to decrease the local N-BP concentration below a BRONJ-triggering threshold.

## Preparation of the HMDP-deformable nanoscale vesicle (DNV) formulation

In our previous proof-of-concept experiment, HMDP directly injected into mouse palatal gingiva in ZOL-pretreated mice prior to the maxillary first molar extraction was shown to prevent the development of BRONJ (*Hokugo et al., 2019*). BRONJ lesions exhibit ulcerative gingival tissue exposing necrotic jawbone. Because injection into pliable oral tissue will be challenging in some cases, and is a more focal delivery, we designed a formulation of HMDP enabling the compound to penetrate through the oral mucosa epithelial layer, making topical delivery to the jawbone possible. Liposomes

have often been used as a drug carrier for controlled delivery to enhance drug concentrations in targeted tissues and to achieve therapeutic effects using minimum drug doses (*Abu Lila and Ishida, 2017*). DNV comprises a modified liposome drug carrier (*Figure 2A*) synthesized by a controlled microfluidics system, which due to nanovesicle deformability allows penetration through the keratinized epithelial layer of skin (*Subbiah et al., 2017*).

HMDP-DNV contains HMDP in the aqueous core and lipid layers formed by a mixture of 1, 2-di oleoyloxy-3-(trimethylammonium) propane-sulfate (DOTAP), dipalmitoylphosphatidylcholine (DPPC), cholesterol (CH), and Span80 (15% v/v) (*Figure 2A*). Span80 is a surfactant that generates DNV deformability. HMDP-DNV was manufactured through a synchronized stringent pipeline for microfluidic synthesis, followed by dialysis and freeze-drying (*Figure 2B*).

DNV containing a far-red fluorophore (Alexa Fluor 647)-labeled ZOL (AF647-ZOL) (*Sun et al., 2016a*) (BioVinc LLC, Pasadena, CA) was manufactured (*Figure 2C*) to examine trans-oral mucosa drug delivery in vivo. We also synthesized 'non-deformable' nanoscale vesicle (nDNV) without Span80 for use as a control.

The BP containing DNV sample was evaluated by particle size and surface zeta potential (*Figure 1D*), which are reported to play important roles in liposome drug delivery behavior (*Rasmussen et al., 2020*; *Bnyan et al., 2018*). The optimal size for this purpose has been shown to be between 100 nm and 200 nm (*Subbiah et al., 2017*). After passing through the gingival and oral mucosa epithelial layers, DNV is expected to deliver HMDP to the bone mineral surface, which is negatively charged in isotonic solution (*Gilbert, 1961*). The binding constants $K_b$ of clinical N-BPs to HAp and their effect when adsorbed at the HAp surface on the zeta potential have been previously investigated (*Nancollas et al., 2006*; *Forte et al., 2017*). At pH 7.4, the zeta potential of HAp is about –4 mV and essentially unchanged by adsorbed ZOL. However, HAp binding etidronate, which is similar in structure to HMDP and like HMDP, lacks a positively charged side chain, showed a negative zeta potential (more negative surface charge). Therefore, cationic DNV with a positive surface potential (between +20 and +40 mV) was selected to target N-BP-chemisorbed jawbone to counteract the latter effect.

The trans-epithelial drug delivery through mouse palatal gingiva was designed to apply HMDP-DNV for 1 hr. During application, the mouse palate was covered by a custom-made oral appliance using dental resin to protect from licking and accidental swallowing (*Figure 2E*). In the initial study, lyophilized AF647-ZOL-DNV was dissolved in MQW at different concentrations (25 μM, 75 μM, and 200 μM) and 3 μl of the AF647-ZOL-DNV solution was applied to the mouse palatal gingival tissue. Forty-eight hours later, euthanized mouse skulls including the palatal alveolar bone were examined for fluorescent signal (*Figure 2F*). AF647 fluorescent signal intensity at the maxillary bone increased with up to an applied concentration of 75 μM of AF647-ZOL and then reached a plateau.

We then prepared AF647-ZOL-DNV and AF647-ZOL-nDNV formulations, which were reconstituted in MQW or 20% polyethylene glycol (PEG). The AF647-ZOL-DNV (75 μM) reconstituted in MQW revealed the highest fluorescent signal, indicating successful AF647-ZOL delivery to the jawbone in vivo (*Figure 2G*). This study also demonstrated that AF647-ZOL-nDNV (75 μM) was less effective in trans-oral mucosa delivery of AF647-ZOL to the jawbone.

## Development of a new mouse model: HMDP-DNV applications prior to dentoalveolar procedures in ZOL-treated mice

Female mice aged 8–10 weeks old were treated by a bolus ZOL intravenous (IV) injection (500 μg/Kg) from the retro-orbital venous plexus. Control mice received a vehicle solution IV injection. One week after the ZOL or vehicle injection, the maxillary left first molar was extracted (*Figure 3A*). Extraction wound healing of the control group was uneventful, and the open gingival wound was mostly closed 1 week after the tooth extraction. By contrast, the tooth extraction wound of the ZOL-treated group was delayed, which was evidenced by a gingival open wound until 4 weeks (*Figure 3B*).

The clinical definition of BRONJ is an unhealed oral wound with the exposed jawbone or the development of fistula reaching to the jawbone surface (*Figure 3C*). AAOMS defined MRONJ including BRONJ as the nonhealing oral wound for 8 weeks in human patients (*Ruggiero et al., 2014*) to differentiate the normal oral wound healing such as tooth extraction, which would be healed within this period. The tooth extraction wound in mice heals in 3 weeks, which is much faster (*Vieira et al., 2015*; *Park et al., 2015*). However, we found that the conventional mouse chaw pellets often delayed the tooth extraction wound healing due to food impaction in the

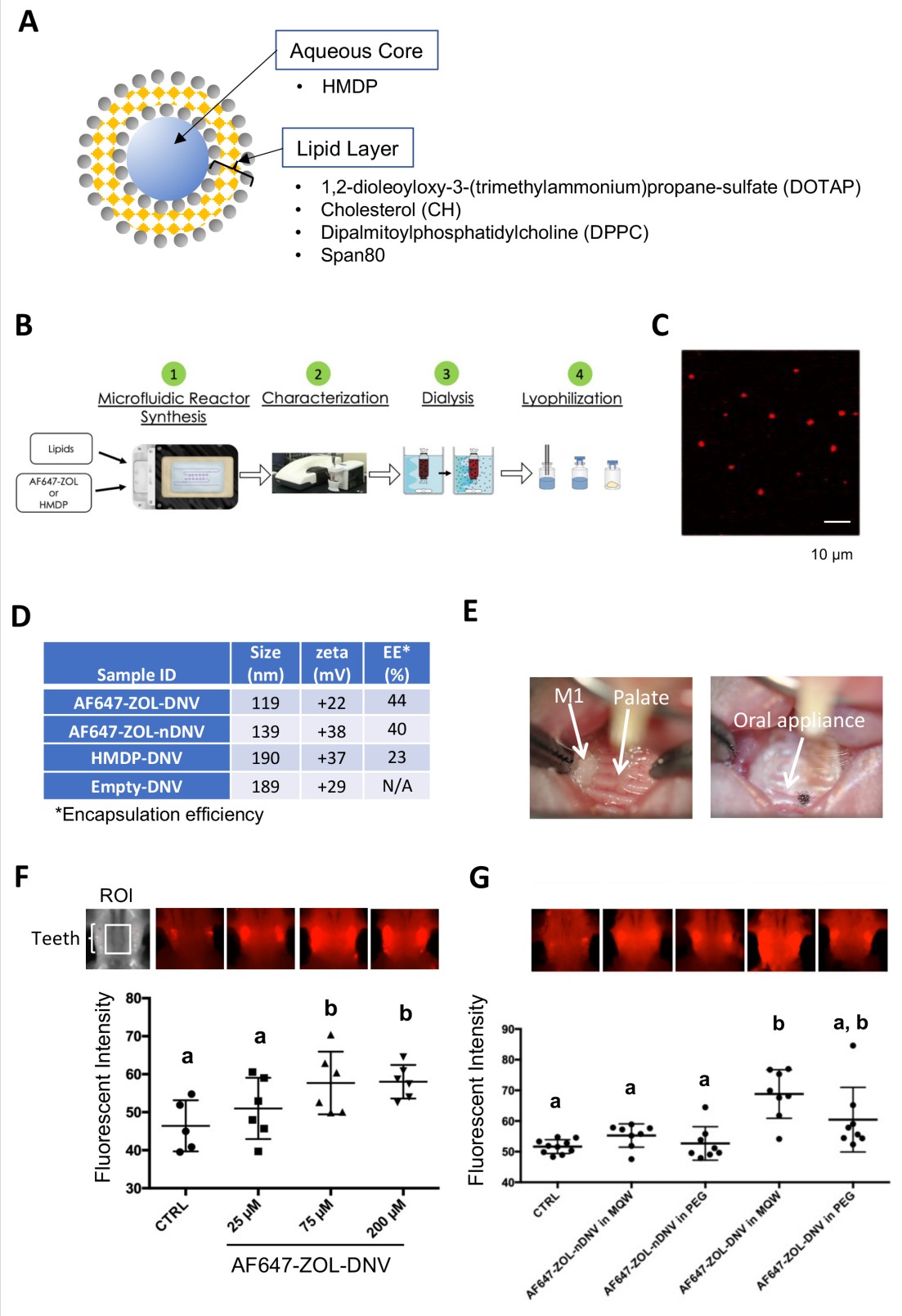

**Figure 2.** Manufacturing and trans-oral mucosal penetration evaluation of hydroxymethylene diphosphonate (HMDP)-DNV and AF647-ZOL-DNV. (**A**) Diagram of deformable nanoscale vesicles (DNV), a liposome derivative. (**B**) Flow diagram of micro-fluidics based DNV synthesis. (**C**) Confocal laser scanning microscopy of AF647-ZOL-DNV. Approximately 100–200 nm DNV particles exhibited an AF647 signal. (**D**) Characterization of DNV formulations. (**E**) Protocol for intra-oral topical application to mouse palatal tissue. The reconstituted DNV solution in MQW (3 µl) was topically applied

*Figure 2 continued*

to the palatal gingiva between maxillary molar teeth and covered by a custom-made oral appliance fabricated with auto-polymerizing dental resin. After 1 hr, the oral appliance was removed. (**F**) After topical application of AF647-ZOL-DNV, mouse maxillary bones were harvested and AF647 fluorescence was measured. The AF647 fluorescent signal from the maxillary bone region of interest (ROI) increased with up to 75 µM AF7647-ZOL in DNV applied and then reached a plateau. (***Figure 2—source data 1***) (**G**) AF647-ZOL-DNV and AF647-ZOL in non-deformable formulation (AF647-ZOL-nDNV) were reconstituted in either MQW or 20% polyethylene glycol (PEG). AF647-ZOL-DNV in MQW most efficiently delivered the drug to the maxillary bone through trans-oral mucosal route. (***Figure 2—source data 2***) In (**F**) and (**G**), the graphs show the mean and SD (n=6 per group), and the Turkey test was used to analyze multiple samples. The statistical significance was determined to be at p<0.05. Different letters (e.g., a, b) are used to show statistically significant differences between multiple groups.

The online version of this article includes the following source data for figure 2:

**Source data 1.** Source data of *Figure 2F*.

**Source data 2.** Source data of *Figure 2G*.

---

extraction socket (***Hokugo et al., 2019***). Refining the mouse tooth extraction model by feeding soft gel diet after tooth extraction for 1 week, we found that the tooth extraction wound was clinically closed as early as 1 week (***Hokugo et al., 2019***). Using this refined mouse tooth extraction model, the BRONJ is assessed as open oral wound for 1 week or longer after tooth extraction. The gross clinical observation is accomplished by using standardized oral photographs. The prevalence of open oral wound is expressed as the percent of animals with unhealed wound in each group at a given healing time.

After tooth extraction, the bony socket undergoes a sequential wound healing resulting in bone regeneration. Micro-CT is a well-established radiographic method suitable for small animal models. We have established a quantitative method to measure the bone volume in the mouse tooth extraction socket using Micro-CT images (***Figure 3—figure supplement 1***). BRONJ delayed the bone regeneration process and often represented as empty tooth extraction socket in ZOL pretreated mice (***Figure 3D***). A recent article reported the diagnostic value of radiographs for BRONJ (***Gaêta-Araujo et al., 2021***), which exhibited radiographic features with unhealed empty extraction sockets and inflammatory lesion (***Figure 3E***).

The development of osteonecrosis is the hallmark of BRONJ histopathology. In the mouse model, the necropsy of harvested maxillary tissues was conducted. The tooth extraction site of ZOL-injected mice exhibited a large area of osteonecrosis defined by a cluster of empty osteocytic lacunae. The gingival connective tissue adjacent to the necrotic bone showed dense inflammatory cell infiltration associated with epithelial hyperplasia (***Figure 3F***). The bone biopsy specimens obtained from human BRONJ patients demonstrated the empty osteocytic lacunae as the definitive sign of osteonecrosis and associated with epithelial hyperplasia (***Figure 3G***), reported as pseudoepitheliomatous hyperplasia (***Zustin et al., 2014***).

Using the mouse BRONJ model, the effect of topical application of HMDP-DNV was examined to determine whether it would modulate the BRONJ symptoms. In this study, Empty-DNV, HMDP in MQW (5 nmol/1.67 mM), or HMDP-DNV in MQW (5 nmol/1.67 mM) were applied topically to the palatal tissue of mice pretreated with ZOL IV injection prior to the maxillary left first molar extraction (***Figure 3H***). One time application of Empty-DNV, HMDP and HMDP-DNV did not affect the delayed tooth extraction wound healing in Micro-CT imaging and necropsy histological analyses (***Figure 3— figure supplement 2***). Therefore, we increased the HMDP dose by twice topical applications prior to the tooth extraction. After 2 × topical applications, the HMDP-DNV treated group showed increased bone regeneration at the equivalent level of the no ZOL-pretreated control group, whereas applications of Empty-DNV and HMDP alone had no significant effect (***Figure 3I***).

Histological examination of the 'HMDP-DNV 2 ×'-treated group showed normal extraction wound healing. The 'Empty-DNV 2 ×' and 'HMDP alone 2 ×'-treated groups revealed extensive alveolar bone osteonecrosis, characteristic of a BRONJ lesion (***Figure 3J***). The area of osteonecrosis in the maxillary alveolar bone in the 'HMDP-DNV 2 ×' application group was significantly reduced (***Figure 3J***). Dentoalveolar surgeries such as tooth extraction have been reported as a risk factor to induce ONJ in patients with a history of N-BP therapy (***Ruggiero et al., 2014***). The results suggest that the twice repeated applications of HMDP-DNV prior to dentoalveolar surgery (tooth extraction) prevented the development of BRONJ.

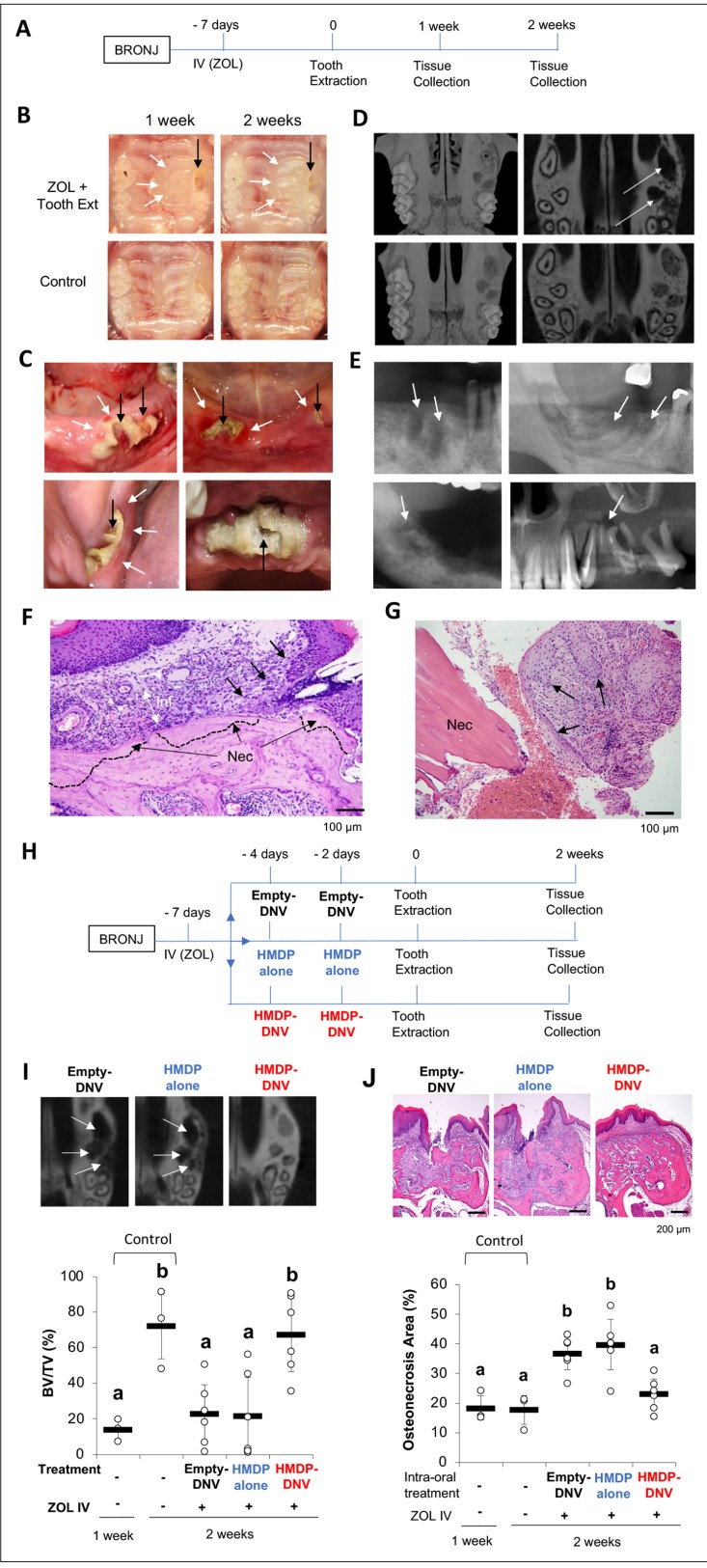

**Figure 3.** Disease phenotypes of mouse and human BRONJ lesion. (**A**) Experimental protocol for inducing a bisphosphonate-related osteonecrosis of the jaw (BRONJ) lesion in mice. Mice received a bolus IV injection of zoledronate (ZOL) (40 nmol, 500 µg/kg) or vehicle saline solution were subjected to maxillary left first molar extraction. (**B**) Intra-oral photographs depicted that mouse pretreated with ZOL IV injection exhibited delayed

*Figure 3 continued on next page*

*Figure 3 continued*

wound healing with sustained open wound (black arrows) and gingival swelling (white arrows). (**C**) Human BRONJ lesions with open tooth extraction wound (black arrows) and gingival swelling (white arrows). (**D**) Micro-CT images of mouse maxilla depicted that delayed bone regeneration in the mesial, buccal and palatal root extraction sockets (white arrows) in ZOL-pretreated mice, a sign of BRONJ symptoms. (**E**) Radiographic demonstration of unhealed tooth extraction of human BRONJ. (**F**) Histological evaluation of mouse BRONJ lesion with osteonecrosis (dotted line; Nec), gingival inflammation (Inf) and epithelial hyperplasia reaching to the necrotic bone (black arrows). (**G**) A biopsy specimen of human BRONJ lesion with osteonecrosis (Nec) and gingival epithelial hyperplasia (black arrows). (**H**) A time course experimental diagram of hydroxymethylene diphosphonate (HMDP)-deformable nanoscale vesicles (DNV) application. All mice received ZOL IV injection. Empty-DNV, HMDP in MQW (HMDP alone) (5 nmol, 3 µl of 1.67 mM) or HMDP-DNV in MQW ( 5 nmol, 3 µl of 1.67 mM) was topically applied to the palatal gingiva prior to the maxillary left first molar extraction. (**I**) Micro-CT evaluation of tooth extraction socket, which remained empty (white arrows) in the groups treated with Empty-DNV or HMDP alone. Two topical applications of HMDP-DNV prior to the tooth extraction significantly increased tooth extraction socket bone regeneration compared to Empty-DNV and HMDP alone. (*Figure 3—source data 1*) (**J**) Histological evaluation. Two topical applications of HMDP-DNV prior to the tooth extraction reduced the development of osteonecrosis compared to the treatment of Empty-DNV and HMDP alone, which remained to exhibit BRONJ phenotype. (*Figure 3—source data 2*) In (**I**) and (**J**), the graphs show the mean and SD (n=3 per untreated control group and n=6 per experimental group), and the Turkey test was used to analyze multiple samples. The statistical significance was determined to be at p<0.05. Different letters (e.g., a, b) are used to show statistically significant differences between multiple groups.

The online version of this article includes the following source data and figure supplement(s) for figure 3:

**Source data 1.** Source data of *Figure 3I*.

**Source data 2.** Source data of *Figure 3J*.

**Figure supplement 1.** Mouse BRONJ model.

**Figure supplement 2.** Single intra-oral topical application of hydroxymethylene diphosphonate (HMDP)-deformable nanoscale vesicles (DNV) to zoledronate (ZOL)-treated mice prior to tooth extraction did not prevent development of a BRONJ lesion.

## Targeted removal of ZOL by HMDP-DNV treatment from nascent BRONJ lesions accelerated disease resolution in mice

BRONJ lesion in mice was defined as unhealed wound 1 week after tooth extraction in ZOL-injected mice, showing delayed gingival wound closure and exposed alveolar bone (*Figure 3B*). Mouse maxillary tissues were harvested 1, 2, and 4 weeks after tooth extraction in the untreated BRONJ group (*Figure 4A*). The incidence of BRONJ lesion sustained at 83.3% and 50.0% at 2 weeks and 4 weeks after tooth extraction (*Figure 4B*). Twice repeated HMDP-DNV topical treatments (two different dose amounts used) were administered to the BRONJ lesion after 1 week since tooth extraction (*Figure 4A*). Intra-oral examination revealed that the gingival wound was clinically closed in all mice at 2 weeks after tooth extraction and also in all mice at 4 weeks after tooth extraction in the topical HMDP-DNV treatment group (*Figure 4B*). The area of open wound (*Figure 4C*) and the area of gingival swelling (*Figure 4D*) were significantly reduced by HMDP-DNV treatment.

Micro-CT imaging demonstrated the impaired bone regeneration in the extraction socket and extended osteolysis in BRONJ mice. By contrast, bone regeneration in the extraction socket was significantly improved by the HMDP-DNV treatment (*Figure 4E*). The osteonecrosis area of maxillary alveolar bone at the tooth extraction site in the untreated ZOL-injected mice progressively increased; however, the HMDP-DNV treated group attenuated the expansion of the osteonecrosis area at 2 weeks of extraction healing, which was significantly decreased at 4 weeks (*Figure 4F*).

Micro-CT analysis of femur bones revealed increased trabecular bone volume and decreased inter-trabecular bone space in the ZOL-injected mice (*Figure 1B and C*), as expected as the antiresorptive effect of ZOL. Intra-oral topical application of HMDP-DNV did not alter the femur bone trabecular architecture, which exhibited typical phenotypes of the ZOL treatment (*Figure 4G*). These results indicate that the treatment efficacy of HMDP-DNV was limited to the jawbone without modulating the antiresorptive effect of legacy ZOL in the other skeletal system.

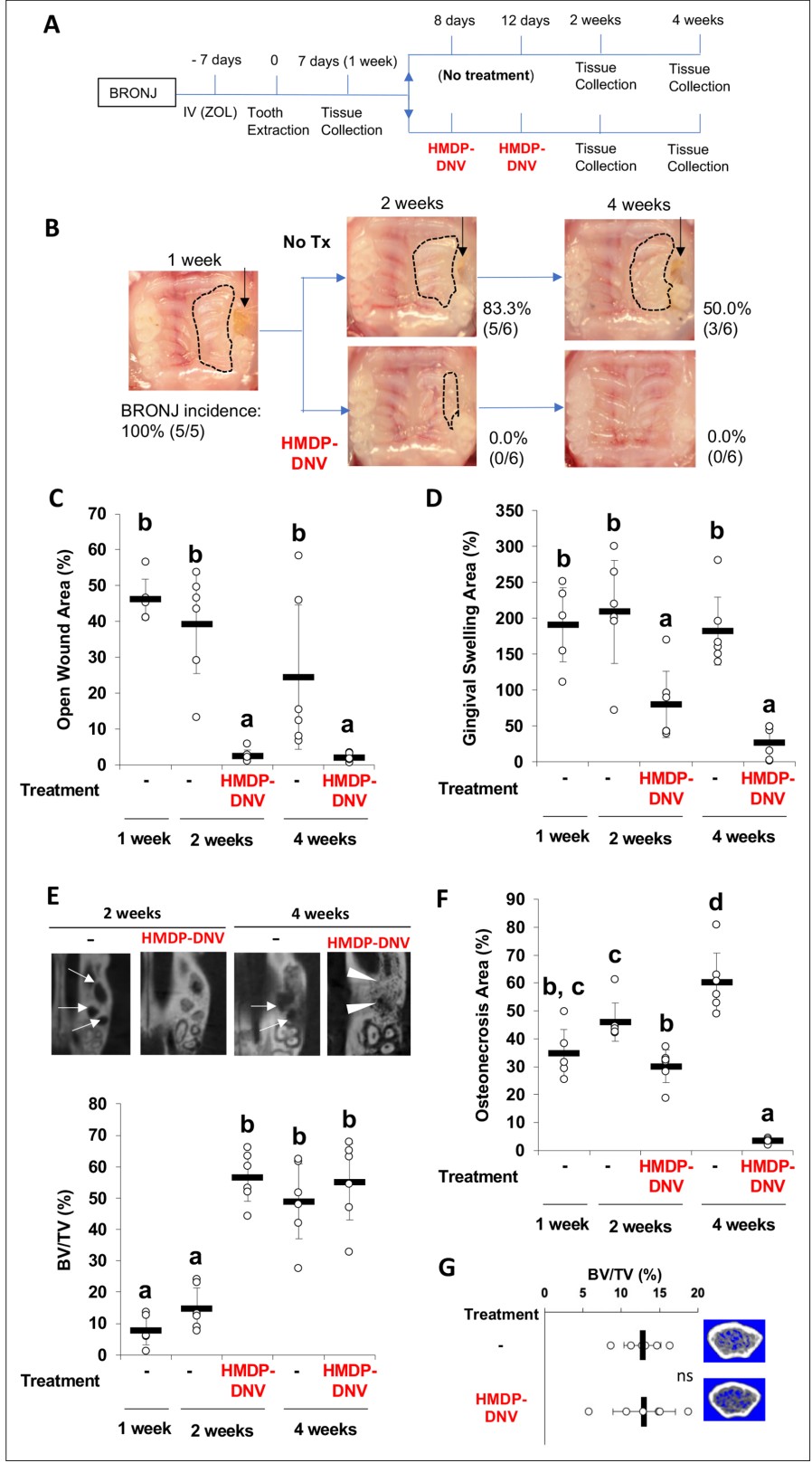

**Figure 4.** Hydroxymethylene diphosphonate (HMDP)-deformable nanoscale vesicles (DNV) topical application to the established BRONJ lesion in mice accelerated the disease resolution. (**A**) The time course experimental protocol. After zoledronate (ZOL) IV injection and tooth extraction, a BRONJ lesion developed, which was treated by two topical applications of HMDP-DNV in MQW at a high dose (15 nmol per application) for the 2-week tissue

*Figure 4 continued on next page*

*Figure 4 continued*

collection experiment and a lower dose (3.8 nmol per application) for the 4-week tissue collection experiment. (**B**) Intra-oral photographs depicted the unhealed tooth extraction wound (black arrows) with gingival swelling (dotted line) in significant percentage of animals in untreated mice group ('No Tx' group) from 2 weeks to 4 weeks after tooth extraction. By contrast, all HMDP-DNV-treated mice exhibited a closed wound in week-2, which was well healed in 4 weeks after tooth extraction. (**C**) The wound opening of untreated mice remained 4 weeks after tooth extraction, while HMDP-DNV topical treatment minimized the wound opening 2 weeks after tooth extraction. (*Figure 4—source data 1*) (**D**) The gingival swelling area was also decreased by HMDP-DNV topical treatment. (*Figure 4—source data 2*) (**E**) Micro-CT analysis showed the delayed bone regeneration in the extraction sockets (white arrows) of untreated ZOL-injected mice. However, HMDP-DNV-treated ZOL-injected mice accelerated the extraction socket bone regeneration, which was further remodeled to generate bone marrow trabecular structure (white arrow heads). The bone volume/total volume of the extraction socket showed the early bone regeneration in the group of HMDP-DNV treatment. (*Figure 4—source data 3*) (**F**) The histological osteonecrosis area progressively increased in the BRONJ lesion. HMDP-DNV topical treatment halted the osteonecrosis area increase at 2 weeks and decreased at 4 weeks after tooth extraction. (*Figure 4—source data 4*) (**G**) The topical application of HMDP-DNV did not affect distant skeletal tissue in femurs. (*Figure 4—source data 5*) In (**C**), (**D**), (**E**), (**F**) and (**G**), the graphs show the mean and SD (n=5–6 per group), and the Turkey test was used to analyze multiple samples. The statistical significance was determined to be at $p<0.05$. Different letters (e.g., a, b, et al.) are used to show statistically significant differences between multiple groups.

The online version of this article includes the following source data for figure 4:

**Source data 1.** Source data of *Figure 4C*.

**Source data 2.** Source data of *Figure 4D*.

**Source data 3.** Source data of *Figure 4E*.

**Source data 4.** Source data of *Figure 4F*.

**Source data 5.** Source data of *Figure 4G*.

## Histological characterization of BRONJ lesion and the effect of HMDP-DNV treatment

The maxillary specimens were decalcified and processed for paraffine embedded histological sections. Histological evaluations of mouse tooth extraction wound showed delayed bone healing and epithelial hyperplasia leading to the exposure of jawbone in untreated ZOL-injected mice. Specimens harvested at 4 weeks after tooth extraction exhibited the development of fistula formed by epithelial hyperplasia reaching to the alveolar bone and a lesion of pustule. More notably, inflammatory cell infiltration to the gingival connective tissue was localized on the necrotic jawbone in the untreated mice and in the empty extraction socket (*Figure 5A*). These histological observations appeared to be consistent with BRONJ lesion in humans. By contrast, the contralateral unwounded side of untreated ZOL-injected mice was free from any noticeable pathology.

The tooth extraction site of the HMDP-DNV treated mice showed the establishment of epithelial continuation and bone regeneration in the bonny socket (*Figure 5A*). It appeared that inflammatory reaction was subsided or less dense than the untreated lesion, and the necrotic jawbone appeared to be actively removed by bone resorption (*Figure 5B*). At 4 weeks after tooth extraction, the tooth extraction wound was seen well healed (*Figure 5A*) and the inflammatory reaction was resolved (*Figure 5B*). Furthermore, these specimens showed the loss of alveolar bone height adjacent to the tooth extraction socket, but the remaining bone contained only a small osteonecrosis area (*Figure 5B*).

It has been well established that tooth extraction induces not only bone regeneration and remodeling by osteoclasts in the bony socket but also a unique set of osteoclasts on the surface of alveolar bone (*Kondo et al., 2022*). Histologically, osteoclasts are distinct as multinuclear large cells adhered to the bone lacunae. In addition, the presence of osteoclasts was confirmed by Cathepsin K (Ctsk) immunohistochemistry (*Littlewood-Evans et al., 1997*). The present study found that Ctsk-positive multinuclear cells were relatively small and flattened in untreated ZOL-injected mice. These osteoclasts were adhered on shallow bone lacunae not only near the tooth extraction socket (Zone A) but also on the alveolar bone surface under the palatine neuro-vascular complex and dense inflammatory cell infiltrate (Zone B) (*Figure 5C*). In some specimens, Ctsk-positive multinuclear cells were found in the gingival connective tissue away from the bone surface.

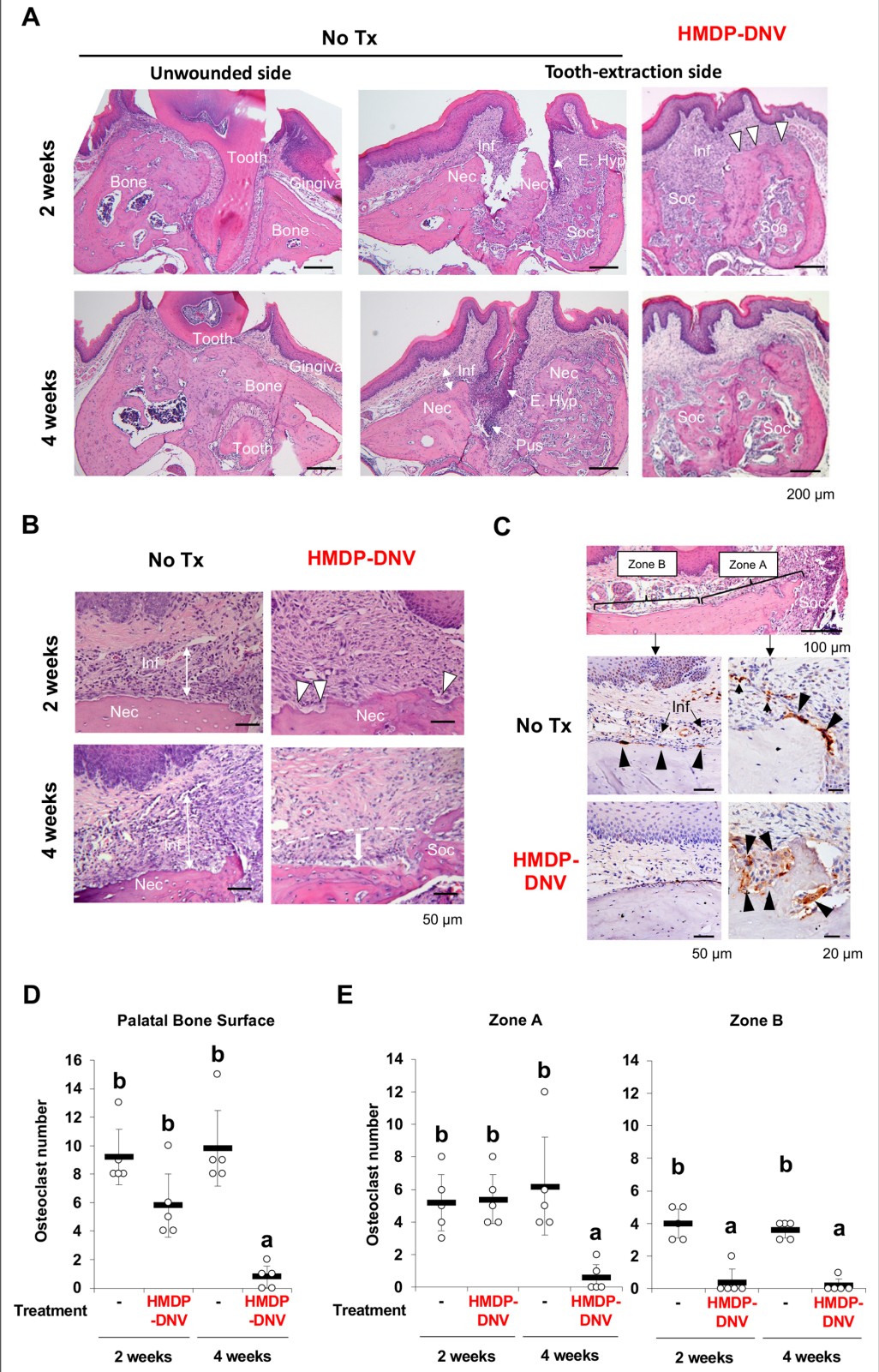

**Figure 5.** Hydroxymethylene diphosphonate (HMDP)-deformable nanoscale vesicles (DNV) topical application normalized tooth extraction wound healing of zoledronate (ZOL)-pretreated mice. (**A**) Histological evaluation depicted BRONJ lesion at the tooth extraction site of the untreated ZOL-injected mice (No Tx) exhibiting that the abnormal epithelial hyperplasia (E. Hyp) extending to the necrotic alveolar bone (Nec) appeared to facilitate

*Figure 5 continued on next page*

*Figure 5 continued*

the sustained open wound, the uneven bone regeneration in the tooth extraction sockets (Soc) and localized infiltration of inflammatory cells (Inf) on the alveolar bone surface. The abnormal wound healing pattern was also observed at 4 weeks after tooth extraction with fistula formation by epithelial hyperplasia reaching to the necrotic bone and a localized pustule lesion (Pus). The unwounded side of ZOL-injected mice did not show any abnormality of remaining tooth (Tooth), alveolar bone (Bone) and overlining gingival tissue (Gingiva). After the HMDP-DNV topical treatment, gingival wound was found closed with more diffused inflammation (Inf) and a sign of bone resorption (white arrowheads) was depicted on the surface of alveola bone. The tooth extraction socket (Soc) showed bone regeneration at week-2, which was further remodeled and matured at week-4. (**B**) High magnification histology of 2 weeks after tooth extraction demonstrated the dense inflammatory cell infiltration (Inf) in the gingival connective tissue and osteonecrosis area (Nec) in untreated ZOL-injected mice (No Tx). The HMDP-DNV treatment attenuated the inflammatory cell infiltration and increased signs of osteoclastic bone resorption (white arrowheads). At 4 weeks after tooth extraction, the untreated mice continued to show dense inflammatory cell infiltration and osteonecrosis. However, mice with the HMDP-DNV treatment showed subsided inflammation and the minimized osteonecrosis area likely due to bone resorption, which appeared to result in alveolar bone loss (dotted white line and arrow) as compared to the regenerated bone in the tooth extraction socket (Soc). (**C**) Cathepsin K (Ctsk) immune-stained osteoclasts on the alveolar bone surface of the untreated ZOL-injected mice were observed not only at the proximal area (Zone A) of the tooth extraction socket (Soc) but also under the palatine neuro-vascular complex (Zone B) with inflammation (Inf). Characteristically, these Ctsk-positive osteoclasts were small and flattened (black arrowheads). In some specimens, Ctsk-positive cells were observed away from the bone surface (small arrowheads). The HMDP-DNV-treated mice showed large Ctsk-positive osteoclasts in deep bone lacunae adjacent to the tooth extraction socket (Zone A). (**D**) Total number of osteoclasts defined as Ctsk-positive multi-nuclear cells on the alveolar bone surface. (*Figure 5—source data 1*) (**E**) Osteoclasts in Zone A and Zone B were separately counted. (*Figure 5—source data 2*) In (**D**) and (**E**), the graphs show the mean and SD (n=5 per group), and the Turkey test was used to analyze multiple samples. The statistical significance was determined to be at p<0.05. Different letters (e.g., a, b) are used to show statistically significant differences between multiple groups.

The online version of this article includes the following source data for figure 5:

**Source data 1.** Source data of *Figure 5D*.

**Source data 2.** Source data of *Figure 5E*.

In contrast, the HMDP-DNV treated group showed large multinuclear Ctsk-positive cells located in the deep bony lacunae. Most osteoclasts in the HMDP-DNV treated group were clustered near the tooth extraction socket (Zone A), whereas only a few osteoclasts were found in Zone B (*Figure 5C*).

The number of osteoclasts was counted on the palatal surface of maxillary alveolar bone. Because the size of maxillary alveolar bone was consistent in the histological sections, the raw osteoclast number was presented (*Figure 5D*). The mean osteoclast number of the untreated ZOL-injected group remained high at 2 weeks and 4 weeks after tooth extraction. The HMDP-DNV treated group showed the progressively decreased osteoclast numbers and at 4 weeks after tooth extraction, almost no osteoclasts were found (*Figure 5D*). The appearance of osteoclasts in Zone A and Zone B was different in the HMDP-DNV treated group that osteoclasts were predominantly found in Zone A (*Figure 5E*).

## Oral barrier immune reaction examined by single cell RNA sequencing

To determine the effect of HMDP-DNV treatment on the oral barrier immunity, we performed the single cell RNA sequencing of dissociated cells from gingival tissue (*Figure 6A*). The gingival oral barrier cells were composed of T lymphocyte and B lymphocyte and myeloid cells (*Figure 6B*). The fraction of T cells in the untreated BRONJ mice contained *Cd8a*+cytotoxic T cells and the upregulated expression of T cell costimulatory molecule CD27 (*Figure 6C*), suggesting the presence of mature CD4+ and CD8+ T cell function (*Hendriks et al., 2000*). Furthermore, a small but distinct expression of *Il17f* indicated the differentiation of highly pro-inflammatory Th17 cells (*Hasiakos et al., 2021*) in the BRONJ lesion. The oral barrier cells from HMDP-DNV treated mice did not show these proinflammatory signatures. Instead, it was noted that the expression of *Foxp3* in their T cell fraction (*Figure 6C*), suggesting the presence of regulatory T cells potentially mediating inflammation resolution (*Campbell and Rudensky, 2020*) and wound healing (*Nosbaum et al., 2016*).

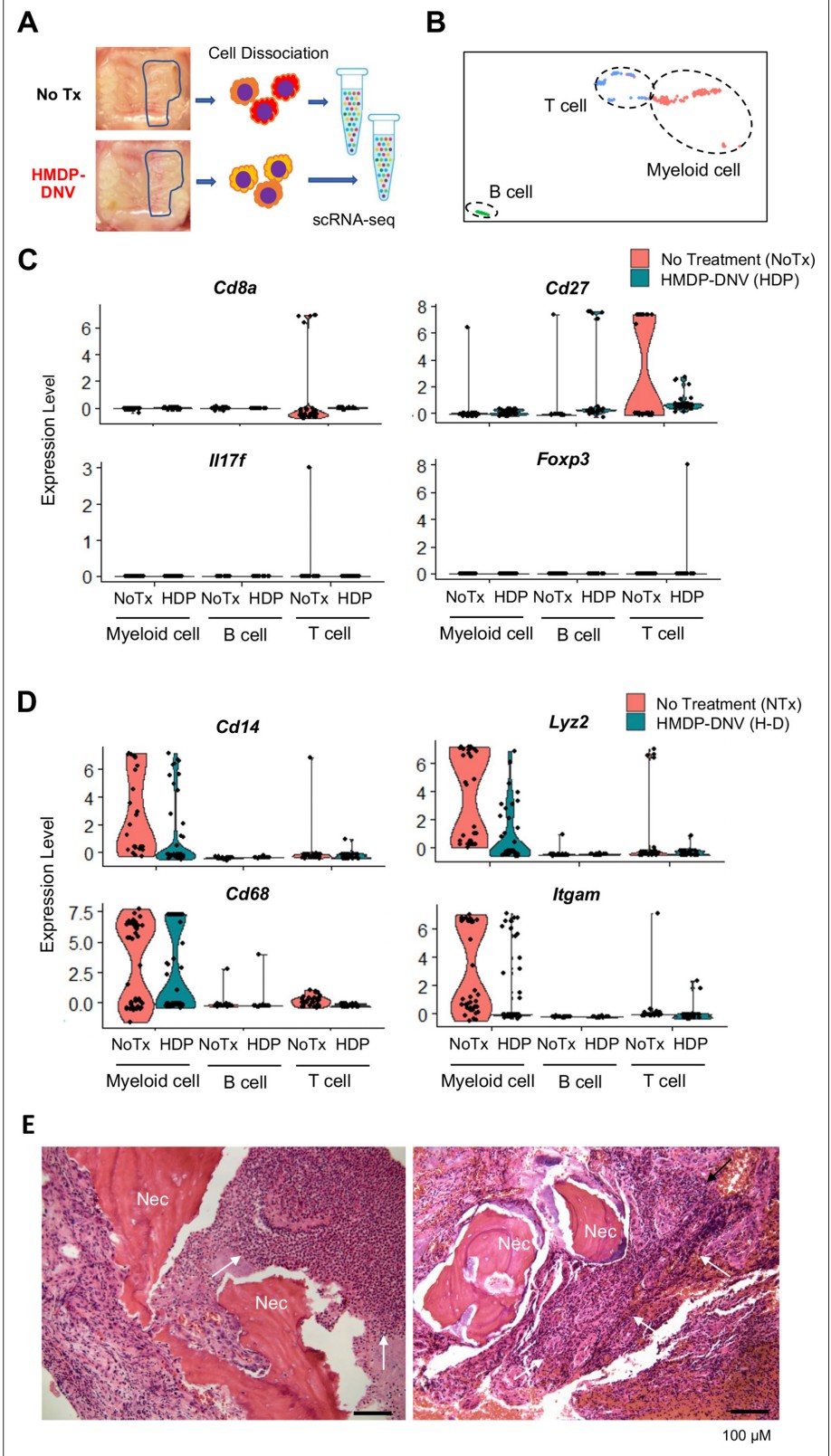

**Figure 6.** Single-cell RNA sequencing of gingival cells of untreated and hydroxymethylene diphosphonate (HMDP)-deformable nanoscale vesicles (DNV)-treated zoledronate (ZOL)-injected mice. (**A**) Two weeks after tooth extraction, gingival tissue adjacent to the tooth extraction site was harvested for cell dissociation followed by single cell RNA-sequencing. (**B**) Using signature gene expression, myeloid cells, T cells, and B cells were identified.

*Figure 6 continued on next page*

*Figure 6 continued*

(**C**) T cell-related gene expression indicated the presence of *Cd8a*+cytototoxic, *Cd27*+matured T cells derived from mouse bisphosphonate-related osteonecrosis of the jaw (BRONJ) gingiva. *Il17f* expression phenotype was decreased by HMDP-DNV treatment, which increased *Foxp3* Treg phenotype. (**D**) Macrophage-related genes demonstrated an increase in M1 macrophages in untreated BRONJ gingiva, which was decreased by HMDP-DNV treatment. (**E**) Human BRONJ biopsy samples showing that necrotic bones (Nec) were associated with a large cluster of neutrophils (white arrows).

Similarly, *Cd14+Lyz2*+myeloid cells in the untreated BRONJ lesion showed the expression of *Cd68* and *Itgam* (*Figure 6D*), suggesting the involvement of macrophages in the innate immune system contributing to the proinflammatory condition (*Chistiakov et al., 2017*). The attenuation or suppression of these proinflammatory functions was suggested in the HMDP-DNV treated oral barrier tissue. The histopathological evaluation of human BRONJ biopsy samples (*Figure 6E*) as well as necropsy specimens from rodent BRONJ models (*Sun et al., 2016b*; *Hokugo et al., 2010*) reported the association with a large cluster of neutrophils.

## HMDP-DNV treatment induced myeloid-derived suppressor cell gene signature in the oral barrier tissue

The myeloid cell fraction of the scRNA-seq data were further divided into neutrophils and macrophages (*Figure 7A*). It was striking that neutrophils in oral barrier tissue of the HMDP-DNV treated group lacked the expression of *TREM1* (*Figure 7B*), which would trigger innate immune activation (*Bouchon et al., 2000*; *Dopheide et al., 2013*). Furthermore, the expression of proinflammatory cytokines, *Il1a*, *Il1b*, and *Tnf* was nearly negated in neutrophils and macrophages by the HMDP-DNV treatment (*Figure 7C*).

It was found that neutrophils as well as macrophages from oral barrier tissue of the HMDP-DNV treated group expressed *Arg1* and *Arg2* (*Figure 7D*), suggesting the stimulation of an anti-inflammatory reaction (*Salybekov et al., 2018*) and tissue regeneration/remodeling (*Colavite et al., 2019*). *Arg*+myeloid cells have also been known as myeloid-derived suppressor cells (MDSC) that negatively regulate the proinflammatory reaction and induce vascular development to support tissue regeneration and wound healing (*Zhou et al., 2018*; *Veglia et al., 2018*). MDSC are diverse population of monocytic and granulocytic myeloid cells that are recently shown to express a common set of signature genes (*Alshetaiwi et al., 2020*). We analyzed the scRNA seq data for the gene signature of MDSC. Macrophage and neutrophil myeloid cells isolated from the oral barrier tissue of the HMDP-DNV treated group expressed all of the evaluated MDSC signature genes, including anti-inflammatory cytokines (*Figure 7D*).

Taken together, the selective removal of ZOL by topical HMDP-DNV treatment from the jawbone prevented the progressive increase of osteonecrosis, normalized osteoclastic activity, and attenuated acute and chronic inflammation, ultimately leading to healing and resolution of the BRONJ lesion (*Figure 7E*).

## Discussion

The pathological mechanism of BRONJ has not been well established. This side effect of N-BPs selectively affects jawbone in the oral cavity. The unique features of the jawbone include its proximity to the oral immune barrier and frequent osteoclastogenesis caused by dentoalveolar infection, inflammation, and wounding (*Moutsopoulos and Konkel, 2018*; *Filleul et al., 2010*). We have hypothesized that the presence of N-BP on the jawbone is a critical causal factor, which interfaces the unique oral environment. To address this hypothesis, the present study applied HMDP to remove the legacy N-BP (*Figure 1*). HMDP is a primary chemical component of Tc-99m-HMDP, a widely used diagnostic agent in single photon emission computed tomography (SPECT) imaging. It is well established for its safety, and the selectivity of HMDP for SPECT imaging of abnormal bone metabolic sites (*Fogelman et al., 1981*) has been demonstrated. A recent clinical report indicated that Tc-99m-HMDP also binds efficiently to MRONJ lesions in humans (*Ogura et al., 2019*). In view of these facts, HMDP was selected and encapsulated in DNVs (HMDP-DNV) to prepare a new topical formulation (*Figure 2*).

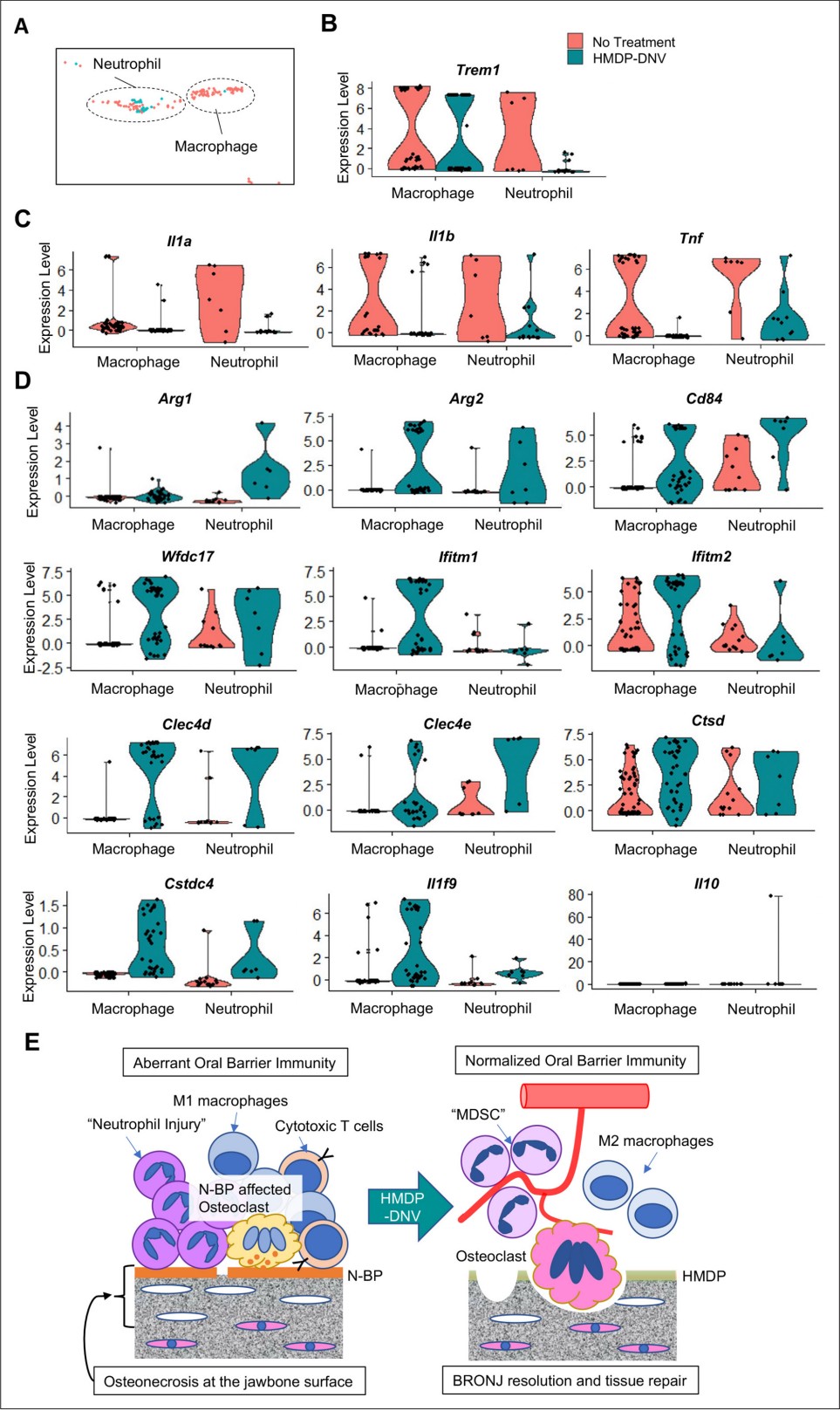

**Figure 7.** Characterization of myeloid immune cells. (**A**) The myeloid cell fraction of scRNA-seq was further divided into neutrophils and macrophages. (**B**) Myeloid immune cells identified by *Trem1* demonstrated a significant decrease of neutrophils by hydroxymethylene diphosphonate (HMDP)-deformable nanoscale vesicles (DNV) treatment. (**C**) Proinflammatory cytokines *Il1a*, *Il1b*, and *Tnf* were highly expressed in macrophages and neutrophils

*Figure 7 continued on next page*

*Figure 7 continued*

from bisphosphonate-related osteonecrosis of the jaw (BRONJ) gingiva. (**D**) Gingival macrophage and neutrophil myeloid cells after HMDP-DNV treatment expressed the multiple signature genes of the myeloid-derived suppressor cell (MDSC): *Arg1, Arg2, CD84, Wfdc17, Ifitm1, Ifitm2, Clec4d, Clec4e, Ctsd, Cstdc4* (*Alshetaiwi et al., 2020*) or M2 macrophage phenotypes *Arg1, Arg2*, as well as anti-inflammatory cytokines *Il1f9* and *Il10*. (**E**) A hypothetical model of BRONJ.

Intra-oral administration of this therapy demonstrated wound closure and radiographic extraction socket bone healing with micro-CT in mice receiving the twice repeated HMDP-DNV topical application and demonstrated a rapid BRONJ resolution (*Figures 3 and 4*). In our mouse model, BRONJ-like lesion was observed only at the tooth extraction area and the contralateral unwounded side did not show any noticeable abnormality, suggesting that the dentoalveolar surgery played an important role in the mouse BRONJ lesion. A recent multicenter retrospective survey confirmed that dentoalveolar surgery and tooth extraction increased the risk of BRONJ (*Hasegawa et al., 2017*).

N-BP adsorbed on bone mineral stays relatively firmly until bone resorption by osteoclasts, and active bone resorption may release N-BP from bone surface to the local environment. Tooth extraction induces not only bone remodeling in the bony socket but also unusual osteoclastogenesis on the external surface of alveolar bone (*Kondo et al., 2022*). The present study demonstrated the oral barrier inflammation of ZOL-pretreated mice, which was densely localized adjacent to the alveolar bone surface (*Figure 5*). It is highly conceivable that tooth extraction-induced osteoclastic activity on the alveolar bone surface could release N-BP to the oral environment. It has been reported that N-BP mediates M1 macrophage polarization (*Zhu et al., 2019*) and impairs neutrophil function (*Kuiper et al., 2012*). The single cell RNA sequencing analysis in the present study suggested the gene expression signature of proinflammatory myeloid cells, which was reversed to anti-inflammatory phenotypes by the HMDP-DNV treatment (*Figures 6 and 7*).

In addition, the present study highlighted the pathological role of released N-BP for prolonged oral inflammation in the BRONJ lesion and the HMDP-DNV treatment induced MDSC phenotype of macrophages and neutrophils (*Figure 7*). ZOL treatment was shown to decrease MDSC in mammary tumor-bearing female mice (*Melani et al., 2007*) and to decrease tumor-associated macrophages (TAM) in mesothelioma-bearing female mice (*Veltman et al., 2010*). MDSC and TAM are important mediators of tumor-induced immunosuppression and the inhibition of MDSC accumulation with N-BP improves the host anti-tumor response in breast cancer (*Melani et al., 2007*) and pancreatic adenocarcinoma (*Porembka et al., 2012*). MDSC has been highlighted in the tumor environment as pathologically activated neutrophils and monocytes with a detrimental role of enhancing tumor growth and metastasis (*Veglia et al., 2021*). The present study demonstrated the clear gene signature of MDSC in the gingival oral barrier tissue after HMDP-DNV-derived removal of ZOL from the jawbone (*Figure 7*). The HMDP-DNV treatment converted from the pro-inflammatory to anti-inflammatory phenotype. The transient presence of M2 macrophages and MDSC during wound healing protects from overreactive immune responses (*Sanchez-Pino et al., 2021*). The reduction of gingival swelling (*Figure 4*) and resolution of chronic gingival inflammation (*Figure 5*) by HMDP-DNV treatment may lead to tissue repair promotion and the re-establishment of homeostasis.

The present study demonstrated that osteoclasts on the alveolar bone surface in ZOL-pretreated mice were found to form abnormally shallow bone resorption lacunae and the HMDP-DNV treatment normalized the osteoclastic activity generating large resorption lacunae (*Figure 5*). The osteonecrosis area of the HMDP-DNV-treated BRONJ mice did not increase compared to the osteonecrosis area before the treatment (*Figure 4*); however, the necrotic bone appeared to be removed by 4 weeks after tooth extraction. The treatment effect of HMDP-DNV was not to revitalize the necrotic bone. The normalized osteoclasts by HMDP-DNV treatment might resorb the necrotic bone. A case series of 25 patients described that the natural resolution of BRONJ was associated with necrotic bone sequestration and debridement leading to primary closure of the exposed bone (*Kim et al., 2016*). It is also possible that HMDP-DNV treatment facilitated the necrotic bone sequestration between 2 weeks and 4 weeks after tooth extraction.

Recently, osteoclasts have been found to maintain the ability to secrete a set of anti-inflammatory cytokines (*Drissi and Sanjay, 2016*; *Tseng et al., 2015*). N-BPs such as ZOL altered osteoclasts to increase the secretion of pro-inflammatory cytokines (*Tseng et al., 2015*). Furthermore, osteoclasts

are found to be an efficient feeder cells supporting natural killer cells in an ex vivo system (*Kaur et al., 2020*). A cluster of osteoclasts induced by dentoalveolar infection and surgery may play a critical role in supporting and regulating the oral barrier immunity (*Tsukasaki et al., 2018*). Denosumab has been used in the same indications as N-BP through a different pharmacological mechanism; but denosumab-related ONJ (DRONJ) demonstrated similar clinical and radiographic symptoms (*Boquete-Castro et al., 2016*). We postulate that the lack of viable osteoclasts as a local immune coordinator may cause the dysregulation of oral barrier immunity leading to the development of BRONJ and DRONJ.

It must be noted that not all patients exposed to antiresorptive medications develop MRONJ. In fact, its incidence is quite small. Therefore, other confounding factors may be involved in the pathogenesis of MRONJ. Clinical case studies proposed an increased risk of BRONJ in patients with autoimmune diseases such as rheumatoid arthritis (*Fujieda et al., 2020*), autoimmune hepatitis (*de Boer et al., 2012*), and Sjogren syndrome (*Kuo et al., 2021*; *Liao et al., 2019*). N-BP-treated rodent models combined with experimental rheumatoid arthritis (*de Molon et al., 2016*) or concurrent dexamethasone treatment (*Inoue et al., 2021*) developed exacerbated BRONJ lesion supporting the susceptibility of compromised systemic immunity. However, a study of larger BRONJ cohort did not show a clear relationship (*Lescaille et al., 2013*). Autoimmune diseases are often treated with immunosuppressive agents such as glucocorticoids, which might affect the susceptibility of BRONJ (*Preidl et al., 2014*; *Schwaneck et al., 2020*). Glucocorticoids are widely used in combination therapies for multiple myeloma and breast cancer involving use of N-BP and other antiresorptive drugs which are implicated in MRONJ (*Inoue et al., 2021*).

The systemic and local confounding factors certainly play a significant role in the disease severity of BRONJ. However, the present studies of topical oral administration of HMDP-DNV demonstrated that targeted removal of ZOL from the jawbone was the single most effective therapy to prevent and accelerate the resolution of BRONJ. We further underline that the local intra-oral application of HMDP-DNV did not affect the distant skeletal system and thus the pathological mechanism of BRONJ is likely localized within the oral tissue. These observations support our hypothesis that being above the threshold dose level of oral N-BP holds the critical key for the pathological mechanism to develop BRONJ. Finally, the work reported here establishes the basis for the development of this novel treatment as an effective prophylactic and therapeutic method for ZOL-induced BRONJ, and possibly for BRONJ associated with other N-BPs.

## Materials and methods
### Chemical reagents
ZOL was acquired from UCLA Medical Center Pharmacy (Reclast, Novartis, Basel, Switzerland). Fluorescent-tagged ZOLs (FAM-ZOL and AF647-ZOL) were obtained from BioVinc LLC (Pasadena, CA). HMDP oxidronic acid; hydroxymethylene-1, 1-bis(phosphonic acid) was acquired from Aroz Technology (Cincinnati, US; Cat # BP-1026) as the disodium salt and characterized by $^1$H NMR (D$_2$O, 600 MHz): δ 3.78 (t, J=16 Hz), $^{31}$P NMR (D$_2$O, 243 MHz): δ 14.90 (s) and by elemental analysis: calculated for CH$_4$O$_7$P$_2$Na$_2$, 4.98% C, 1.92% H; found 4.97% C, 1.91% H (>99% purity). DOTAP (1, 2-dioleoyloxy-3-[trimethylammonium]propane-sulfate), DPPC (dipalmitoylphosphatidylcholine), CH (cholesterol) and the nonionic surfactant Span80 were acquired from Sigma-Aldrich (St. Louis, MO).

### The effect of ZOL and HMDP in mouse femur bone structure in vivo
Female C57BL/6J mice (n=5 per group) were anesthetized by isoflurane inhalation and 100 µl of ZOL (40 nmol), HMDP (40 nmol) or saline vehicle solution was injected to retro-orbital venous plexus (*Park et al., 2015*; *Sun et al., 2016b*). Three weeks after the injection, mice were euthanized and femur bones were harvested for micro-CT imaging (µCT40, Scanco Medical AG, Southeastern, PA) following the standard procedure. Bone parameters were determined using the proprietary analysis program.

### Competitive displacement removal of ZOL by HMDP in vitro
Cell culture wells coated with carbonate apatite (Bone resorption assay plate 24, Cosmo Bio Co. Ltd, Tokyo, Japan) were incubated with fluorescent-tagged ZOL (10 µM FAM-ZOL) overnight at 37°C, 2% CO$_2$, followed by three washes with Milli-Q treated pure water (MQW) for 10 min each. The FAM-ZOL

coated wells were then incubated with 10 µM HMDP in MQW (n=6) or MQW (N=3) for 2 hr at 37°C, 2% $CO_2$ followed by three washes with MQW. One group of HMDP treated wells (n=3) were treated by the second application of 10 µM HMDP. Other wells were treated by MQW. After washes, the FAM fluorescent signal of each well was evaluated (IVIS Lumina II, PerkinElmer, Waltham, MA): excitation 465 nm; and emission filter: GFP (*Sun et al., 2016a*). The region of interest was set to the well size and the fluorescent signal was measured. Separately, all wash solutions were subjected to fluorescent signal evaluation.

## Osteoclast resorption pit formation assay in vitro

Cell culture wells coated with carbonate apatite were incubated with ZOL (10 µM in MQW) for overnight at 37°C, 2% $CO_2$ followed by extensive washes. ZOL-preincubated wells were treated with 10 µM HMDP once (n=3) or twice (n=3) as described above. Control wells were treated with MQW (n=3). After the final wash, culture medium (MEM containing 10% fetal bovine serum and 1% antibiotics mix) was added to all wells. Then, RAW264.7 cells ($2.5×10^4$ cells per well) were inoculated to each well in culture medium supplemented by mouse recombinant receptor activator of nuclear kappa-B ligand (RANKL; Sigma-Aldrich) (100 ng/ml) and incubated at 37°C, 2% $CO_2$. The culture medium was changed after 3 days. The resorption pits were photographed after 6 days of incubation after the cells were removed with 0.25% Trypsin. The total area of resorption pit was measured using a Java-based image processing program (ImageJ, NIH, Bethesda, MD).

## Synthesis of HMDP-DNV and AF647-ZOL-DNV

Fabrication of the DNV formulation containing BP compounds followed the published method (*Subbiah et al., 2017*). Briefly, cationic DOTAP lipid with DPPC and CH were dissolved in a 10 mM chloroform solution at a 5:3:2 volume ratio and the mixture was allowed to dryness. The dried lipid mixture was resuspended in isopropyl alcohol (final concentration 10 mM). To this solution, Span80 (15% v/v) was added. In the case of non-deformable DNV (nDNV), Span80 was not added. The lipid solution was filtered (0.2 µm) before injection in the reactor.

The aqueous solution was comprised of deionized water and the BP compound (12.9 µM) and filtered (0.2 µm). The organic and aqueous solutions were used for DNV synthesis using microfluidics reactor (Syrris, Royston, UK) with a 26 µL reactor chip at 1000 µL/min organic stream and 5000 µL/min aqueous stream, followed by dialysis to remove the non-encapsulated BP compound and free lipids and then freeze-dry processes. Physical characteristics such as the size and zeta potential were determined by a Malvern Zetasizer (Nano-ZS: Malvern, Worcestershire, UK) following the manufacturer's protocol.

BP encapsulation efficiency (EE) was calculated as percent of AF647-ZOL amount in the DNV relative to the initial AF647-ZOL amount used in the synthesis. An aliquot of the lyophilized AF647-ZOL-DNV or AF647-ZOL-nDNV was lysed by adding acetonitrile (ACN). Next, samples were shaken for half an hour, then 350 µL of water was added to solubilize the compounds and samples were centrifuged for 10 min at 22,000×g. The supernatant was collected for HPLC. A standard curve was created using the original 12.9 µM stock solution of each of the compounds by HPLC for AF647-ZOL-DNV. Using the equation for the line of best fit for the signals received, we inputted the average signal given by the samples post-dialysis to obtain their concentration. It should be noted that the amount/concentration of all BP-DNV or BP-nDNV discussed in the following in vitro and in vivo studies were all calculated based on the encapsulated BPs in the DNV/nDNV formulation.

## LCMS analysis method for the encapsulation efficiency measurement of HMDP-DNV: sample preparation

Five hundred (500) µL of ice-cold LCMS grade ACN was added to vial containing lyophilized HMDP-DNVs. The mixture was vortexed for 3 mins and sonicated for 20 mins to break the DNVs. This sequence was repeated once. Next, 2 mL LCMS grade water was added to each vial and the mixture was vortexed for 10 s and centrifuged at 10,000 rpm for 30 mins. Following the centrifugation, 1.5 mL of the supernatant was collected and separated from the pellet. A 5 µg/mL HMDP stock solution was prepared from the supernatant using maximum theoretical encapsulation value of 100 µg/vial. From the stock, 200 µL, 100 µL, and 40 µL corresponding to 1000 ng, 500 ng, and 200 ng HMDP respectively, were added to 1-dram glass vials. To each vial was added 200 µL of 500 ng/mL etidronate

disodium solution as internal standard (IS). The samples were lyophilized. 500 µL trimethyl ortho-acetate (TMOA) and 150 µL acetic acid was added to the lyophilized samples, and the mixture was heated at 100°C for 1 hr for derivatization. The solvents were evaporated under a stream of air. The residue was dissolved in 1 mL LCMS water. The samples were filtered, transferred to 1 mL HPLC vials, and the HMDP content was analyzed using LCMS. For generating standard curve, standard solutions were prepared by adding known concentrations of HMDP to 1-dram glass vials along with 200 µL of 500 ng/mL etidronate disodium as IS. The solutions were lyophilized, derivatized, and processed as mentioned above.

The LCMS system consisted of Waters Aquity H-class UPLC coupled with Waters Xevo G2-XS QTOF equipped with Masslynx 4.2 software (Waters, Massachusetts, USA). Chromatography was performed on Syncronis C18 Column (2.1×50 mm, particle size 1.7 µm, Thermo Scientific) using water (mobile phase A) and ACN (mobile phase B) at 30 °C and 0.4 mL/min. The chromatography gradient was 0–0.3 mins (0% B), 0.3–0.8 mins (0–100% B), 0.8–1.6 mins (100% B), 1.6–2 mins (100–0% B), and 2–3.5 mins (0% B). The QTOF was operated in positive ion mode monitoring $m/z$ transitions for methylated derivates of HMDP (313->161) and etidronate (327->267). Measurements of at least three different concentrations with 3×–5× replicating tests at each concentration were performed, and the final EE value is calculated as the average value of these tests. Using the theoretical concentration of the samples after synthesis, we were able to calculate the encapsulation efficiency of the liposomes as 23%, which was verified (+/-5%) by direct analysis of the unmodified HMDP-DNV formulation by a $^{31}P$ NMR method.

## Estimation of HMDP content in HMDP-DNV by $^{31}P$ NMR

To verify the HMDP content of HMDP-DNV, we conducted an independent analysis based on $^{31}P$ NMR. Using the LCMS-based approach, the encapsulation efficiency (EE) was calculated as percent of HMDP amount in the DNV relative to the initial HMDP amount used in the synthesis. By this method, the final EE value for HMDP-DNV averaged to 23% which corresponds to a concentration of 3.0 µM for HMDP encapsulated by the liposome. Unlike the LCMS-based approach, $^{31}P$ NMR analysis does not require chemical derivatization of the HMDP. Thus, an aliquot of the HMDP-DNV formulation was used directly as the sample. The average integration of the $^{31}P$ NMR signal for an accurately weighed sample of HMDP in $D_2O$ was determined relative to the signal of an external reference of known concentration (analytically pure MDP [methylenebisphosphonic acid, disodium salt]). The same system was then used to determine the concentration of HMDP in the HMDP-DNV formulation, in $D_2O$. The value obtained by the latter method was 2.7 µM, which agreed within error with the value of 3.0 µM from the LCMS analysis.

## Topical drug application to the mouse palatal mucosa and AF647-ZOL-DNV delivery to the jawbone

We designed a topical application protocol for delivery of BP-DNV to the maxillary jawbone through palatal mucosa. Mouse oral appliances were fabricated using clear orthodontic dental resin to fit to the entire mouse palatal mucosa surface between the molar teeth. Mice were anesthetized by isoflurane inhalation and placed on a surgical bed to open the mandible. Threeµl of AF647-ZOL-DNV or AF647-ZOL-nDNV reconstituted in MQW or 20% polyethylene glycol solution was pipetted to the palatal mucosa and covered by an oral appliance. The mandible was closed with a bite block and the animal was placed in the anesthesia chamber. Mice were kept anesthetized for 1 hr and the oral appliance was then removed. After 24–48 hr, mouse maxillae were harvested and the AF647 fluorescent signal was measured by an imaging reader (LAS-3000, Fujifilm Corp, Tokyo, Japan). While AF647-ZOL-DNV assisted the trans-oral epithelial delivery, some AF647-ZOL-DNV appeared to be disintegrated in the oral cavity and the released AF647-ZOL was adsorbed to the exposed tooth surface or the enamel structure. Thus, we set the region of interest (ROI) only on the palatal bone excluding the tooth structure.

## Induction and characterization of BRONJ lesion in the mouse model

Female C57Bl/6J mice (8–10 weeks of age) received a bolus intravenous injection of ZOL (500 µg/kg in saline solution) or vehicle saline solution from the retro-orbital venous plexus. It was important to prevent traumatic tooth extraction, which caused a serious confounding factor for sound tooth

extraction wound healing. The younger mice at 8–10 weeks allowed consistent atraumatic tooth extraction in our study (*Park et al., 2015*; *Sun et al., 2016b*). Oneweek later, the maxillary left first molar was extracted under general anesthesia by isoflurane inhalation (*Park et al., 2015*; *Sun et al., 2016b*). After tooth extraction, all mice were fed gel food (DietGel Recovery, Clear $H_2O$, Portland, ME) for 1 week and then switched to regular mouse pellet chaw. After switching to the regular pellet chaw, all mice were briefly anesthetized once a week and food and debris impaction was removed. Mice were euthanized at 1 week, 2 weeks, or 4 weeks after tooth extraction and femur bones and the maxillary tissues including the extraction wound were harvested after being photographed. The buffered formalin-fixed maxillary tissues and femur bones underwent Micro-CT imaging. The maxillary tissues were demineralized by 1 M EDTA in 4°C and prepared for paraffin-embedded histological sections stained with hematoxylin and eosin. Digitized histological images were examined, and the area of osteonecrosis was measured using the ImageJ program. The mouse oral lesion, Micro-CT radiography and histopathology were compared to clinical information from human BRONJ patients.

## Topical application of HMDP-DNV to the mouse BRONJ model prior to tooth extraction

ZOL-injected mice were treated with topical application of 3 µl of Empty-DNV, HMDP in MQW (5 nmol/1.67 mM) or HMDP-DNV in MQW (5 nmol/1.67 mM) to the palatal oral mucosa. The topical formulation was applied 1 time (Figure S3A) or 2 times (*Figure 3H*) in a week and then the maxillary left first molar was extracted. Twoweeks after tooth extraction, mouse maxillae and femur bones were harvested. The development of BRONJ lesion was examined as described above.

## Topical application of HMDP-DNV to the established mouse BRONJ lesion

ZOL-injected mice underwent a maxillary left first molar extraction. Delayed healing and BRONJ-like jawbone exposure were confirmed 1 week after tooth extraction. HMDP-DNV in 3 µl MQW (high dose (15 nmol of HMDP-DNV per treatment) was used for the 2-week tissue collection experiment; lower dose (3.8 nmol of HMDP-DNV per treatment) was used for the 4-week tissue collection experiment) was topically applied to the BRONJ lesion and surrounding gingival tissue. This treatment was applied 2 times total in a week. The maxillary tissues and femur bones were harvested at 2 weeks and 4 weeks after tooth extraction and the BRONJ lesion was characterized as described above.

## Histological characterization of osteoclasts

In the histological sections stained by hematoxylin and eosin (H&E), osteoclasts were identified as cells containing at least three nuclei and localized on the bone surface. Selected histological sections were further subjected to immunohistochemical staining with anti-cathepsin K (Ctsk) antibody (Cat# ab19027, Abcam, Waltham MA). Sections were deparaffinized and treated with an antigen retrieval procedure using citrate buffer (pH 6.0) and high heat for 2 min followed by a conventional blocking process. After the incubation with the anti-Ctsk rabbit polyclonal primary antibody, the anti-rabbit IgG secondary antibody was used with 3, 3'-diaminobenzine chromogenic reaction. The sections were then counter stained with hematoxylin.

Using both H&E stained and Ctsk immunohistochemical sections, the number of osteoclasts was counted on the palatal surface of maxillary alveolar bone of the tooth extraction side. When the gingival connective tissue was significantly pealed from the bone surface, the sections were excluded. Next, the palatal surface of maxillary bone was equally divided into 2 zones: Zone A was the lateral zone toward the edge of tooth extraction socket; and Zone B was the medial zone toward the mid-palatine suture, typically under the palatal neuro-vascular structure of gingiva. Because the mouse maxillary bone was highly consistent for the palatal surface size, the osteoclast number was not normalized by the bone surface area and was presented as raw data.

## Single cell RNA sequencing of gingival oral barrier immune cells

After the HMDP-DNV treatment of the BRONJ mice, the gingival tissue from the tooth extraction side of the palate was harvested (n=4). The gingival tissue of untreated BRONJ mice was also harvested (n=4). Gingival tissues were cut into 1 mm pieces and placed in digestion buffer containing 1 mg/ml collagenase II (Life Technologies, Grand Island, NY), 10 units/ml DNase I (Sigma-Aldrich) and 1%

bovine serum albumin (BSA; Sigma-Aldrich) in Dulbecco's modified Eagle's medium (DMEM; Life Technologies) for 20 min at 37°C on a 150 rpm shaker. The tissues were passed through a 70 μm cell strainer. The collected cells were pelleted at 1500 rpm for 10 min at 4°C and resuspended in phosphate buffered saline (PBS; Life Technologies) that was supplemented with 0.04% BSA (Cell suspension A).

The remaining tissues were further incubated in 0.25% trypsin (Life Technologies) and 10 units/ml DNase I for 30 min at 37°C on a 150 rpm shaker. The combined trypsin-released cells and collagenase II-released cells were counted and the equal number of cells (2000~3000) from each animal were combined in the group for single cell RNA sequencing (10 × Genomics, San Francisco, CA). The Cell Ranger output of single cell RNA sequencing data were analyzed using R-program (Seurat, https://satijalab.org/seurat/).

## Statistical analysis

The mean and SD were used to describe the data. The Turkey test was used to analyze multiple samples and statistical significance was considered to be achieved if $p < 0.05$.

## Study approval

All animal experiments were performed at UCLA. All the protocols for animal experiments were approved by the UCLA Animal Research Committee (ARC# 1997–136) and followed the Public Health Service Policy for the Humane Care and Use of Laboratory Animals and the UCLA Animal Care and Use Training Manual guidelines. The C57Bl/6J mice (Jackson Laboratory) were used in this study. Animals consumed gel or regular food for rodents and water ad libitum and were maintained in regular housing conditions with a 12 hr light/dark cycles at the Division of Laboratory Animal Medicine at UCLA.

The human BRONJ biopsy data were collected under the approval by the UCLA Institutional Review Board (IRB #17–000721). Clinical demonstration of human BRONJ was obtained from patients of UCLA School of Dentistry clinic with the general consent for educational use. The information was not part of investigator-initiated research.

# Acknowledgements

We thank Dr. Hodaka Sasaki, UCLA School of Dentistry and Tokyo Dental College for designing the mouse oral appliance, Dr. Jimmy Hu of UCLA School of Dentistry for his guidance in single cell RNA sequence analysis, the UCLA Translational Pathology Core Laboratory for histological specimen preparation, the UCLA Technology Center for Genomics & Bioinformatics for 10XGenomics single cell RNA sequencing, and Inah Kang, USC for skilled assistance in preparing the manuscript. National Institutes of Health grant R01DE022552 (IN). National Institutes of Health grant R44DE025524 (FHE, IN). Bridge Institute at USC, Drug Discovery Center (CEM). Tohoku University Leading Young Researcher Overseas Visit Program Fellowship (HO). Japan Society for the Promotion of Science Research Fellowship for Young Scientists 19J117670 (TK). National Institutes of Health grant C06RR014529 (UCLA Facility).

# Additional information

### Competing interests

Philip Cherian: is an employee in BioVinc LLC. Frank H Ebetino: holds equity in BioVinc LLC and is an employee and has executive management positions in BioVinc LLC. Shuting Sun: holds equity in, is an employee of, and has executive management position in BioVinc LLC. Charles E McKenna: is a board member and holds equity in BioVinc LLC. Ichiro Nishimura: was a consultant of BioVinc LLC. The other authors declare that no competing interests exist.

## Funding

| Funder | Grant reference number | Author |
| --- | --- | --- |
| National Institute of Dental and Craniofacial Research | R01DE022552 | Ichiro Nishimura |
| National Institute of Dental and Craniofacial Research | R44DE025524 | Frank H Ebetino |
| Tohoku University | Leading young researcher overseas visit program fellowship | Hiroko Okawa |
| Japan Society for the Promotion of Science | 19J117670 | Takeru Kondo |
| National Center for Research Resources | C06RR014529 | Ichiro Nishimura |
| BioVinc LLC | Gift fund | Ichiro Nishimura |

The funders had no role in study design, data collection and interpretation, or the decision to submit the work for publication.

## Author contributions

Hiroko Okawa, Takeru Kondo, Formal analysis, Funding acquisition, Investigation, Methodology; Akishige Hokugo, Conceptualization, Formal analysis, Investigation, Methodology, Writing – review and editing; Philip Cherian, Formal analysis, Investigation, Methodology; Jesus J Campagna, Nicholas A Lentini, Investigation; Eric C Sung, Yi-Ling Lin, Data curation; Samantha Chiang, Data curation, Writing – review and editing; Frank H Ebetino, Conceptualization, Data curation, Funding acquisition, Writing – review and editing; Varghese John, Supervision, Investigation; Shuting Sun, Conceptualization, Data curation, Writing – review and editing; Charles E McKenna, Conceptualization, Funding acquisition, Investigation, Methodology, Writing – review and editing; Ichiro Nishimura, Conceptualization, Formal analysis, Supervision, Funding acquisition, Investigation, Methodology, Writing – original draft, Project administration, Writing – review and editing

## Author ORCIDs

Akishige Hokugo (iD) http://orcid.org/0000-0002-7097-3364
Nicholas A Lentini (iD) http://orcid.org/0000-0002-1701-0753
Shuting Sun (iD) http://orcid.org/0000-0002-3635-9029
Charles E McKenna (iD) http://orcid.org/0000-0002-3540-6663
Ichiro Nishimura (iD) http://orcid.org/0000-0002-3749-9445

## Ethics

Human subjects: This study was not conducted on human subjects. However, the manuscript contains a clinical demonstration of human BRONJ obtained from patients of UCLA School of Dentistry clinic with a general consent for educational use. The information was not part of investigator-initiated research.

All animal experiments were performed at UCLA. All the protocols for animal experiments were approved by the UCLA Animal Research Committee (ARC# 1997-136) and followed the Public Health Service Policy for the Humane Care and Use of Laboratory Animals and the UCLA Animal Care and Use Training Manual guidelines. The C57Bl/6J mice (Jackson Laboratory) were used in this study. Animals consumed gel or regular food for rodents and water ad libitum and were maintained in regular housing conditions with a 12-hour-light/dark cycles at the Division of Laboratory Animal Medicine at UCLA.

## Decision letter and Author response

Decision letter https://doi.org/10.7554/eLife.76207.sa1
Author response https://doi.org/10.7554/eLife.76207.sa2

## Additional files

### Supplementary files
• Transparent reporting form

### Data availability
All data generated or analyzed during this study are included in the manuscript and supporting file. Single cell RNA sequencing data have been deposited in GEO under accession code GSE193110.

The following dataset was generated:

| Author(s) | Year | Dataset title | Dataset URL | Database and Identifier |
|---|---|---|---|---|
| Kondo T, Nishimura I | 2022 | Single Cell RNA sequencing of BRONJ disease control and HMDP-treated mouse gingiva | https://www.ncbi.nlm.nih.gov/geo/query/acc.cgi?acc=GSE193110 | NCBI Gene Expression Omnibus, GSE193110 |

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
