## [Editor Report]

The manuscript shows that bisphosphonate-related osteonecrosis of the jaw, a rare complication of osteoporosis treatment and bone marrow cancers, was prevented/alleviated in mice using a novel treatment which works by reversing the associated oral inflammation. The work in this manuscript is valuable and will be of significant interest to investigators in the bone and dental fields who conduct pre-clinical research.

---

## [Decision Letter]

**Decision letter after peer review:**

Thank you for submitting your article "Mechanism of bisphosphonate-related osteonecrosis of the jaw (BRONJ) revealed by targeted removal of legacy bisphosphonate from jawbone using competing inert hydroxymethylene diphosphonate" for consideration by *eLife*. Your article has been reviewed by 2 peer reviewers, one of whom is a member of our Board of Reviewing Editors, and the evaluation has been overseen by Mone Zaidi as the Senior Editor. The reviewers have opted to remain anonymous.

Essential revisions:

(1) Data from a longer study period should be provided.

(2) More than one time point should be provided.

(3) Bone resorption endpoints should be measured.

*Reviewer #1 (Recommendations for the authors):*

A minor weakness is that BRONJ can occur clinically in the setting of suppressed inflammation with the use of glucocorticoids- arguing against a purely pro-inflammatory mechanism.

Given that BRONJ usually happens in the setting of high frequency doses of BPs, the authors could assess whether HMDP-DNV displacement works in that setting too, which is probably the most clinically relevant one.

Finally, it would be interesting to note if there is evidence for pro-inflammatory signaling at other mucosal sites in the mice with BP treatment.

*Reviewer #2 (Recommendations for the authors):*

Specific concerns and suggestions for improvement are given below:

– A major weakness of this study is that it only investigates the effects of treatments 2 wks post-extraction. The ONJ-related exposed bone is a chronic manifestation of antiresorptive medication. In clinical studies, as the authors mentioned, the efficacy of treatments to treat BRONJ related oral wounds takes a long time, even years.

– The authors justified the evaluation of the HMDP-DNV treatment for only 2 wks based on the experience that providing a soft diet post-extraction contributes to the oral wound's healing process; however, as it is well-established, the uneventful wound healing (without infection) of the extraction socket in a rodent could take several weeks and even months to complete if bone healing is taken into account. Moreover, if we consider that N-BPs delay healing, an experiment that only investigates the model for two weeks is insufficient; therefore, it is likely that the model is investigating the effects on early healing but not BRONJ.

– Another limitation of the study is that it only tested a one-time point. If the treatment effect of HMDP-DNV, as the authors implied, were not to revitalize the necrotic bone but to remove the affected bone by normalizing the osteoclastic activity and halting further necrosis, earlier time points would be needed to substantiate these statements. In addition, to investigate BRONJ development and progression later time points would have been important.

– The authors stated that BRONJ is predominantly induced by the oral N-BP and suggested that HMDP-DNV is a possible effective therapy for BRONJ. Then, what would be the explanation that only 0.005 % and 1-5% of the patients taking osteoporosis or oncologic doses of N-BPs get BRONJ but not all of them?

– Since bisphosphonates' main effects inhibit osteoclast and bone resorption, it is surprising that osteoclasts and bone resorption were not quantitatively assessed in the alveolar bone tissues in the in vivo study.

– In the introduction, the authors presented an alternative MRONJ "prevalence" estimation of approximately 36,000 new cases per year (Table 1). They claimed that still considered within the FDA's orphan disease definition of under 200,000 annually. There are two possible issues with this sentence. First, the definition states 200,000 people, BUT the estimation is not per year but at a specific time. Second, it is incorrect to refer to prevalence. Prevalence is the proportion of people in a population with a particular disease or condition. Therefore, it is more appropriate to call it incidence since it refers to "new cases" of disease in a population over a specific period.

– In the introduction, the authors claimed that HMDP-DNV topical applications to "nascent mouse BRONJ lesions" resulted in accelerated gingival wound closure and bone socket healing as well as attenuation of osteonecrosis development. This statement is inaccurate because the BRONJ lesion never developed in the HMDP-DNV treated group, according to the data presented at 2 wks. Therefore, it appears to be more a "preventive approach." In order to make this statement, earlier time points would be required to substantiate the "nascent BRONJ lesions" developed in HMDP-DNV.

– Please add a reference for the FRAX score (line 62-64).

– Refs 5 and 9 are only for osteoporosis doses (lines 66-67) but not the oncologic dose in the abstract.

– Considering that BRONJ affects mature adults, why did the authors utilize 8-10 wks old mice? The wound healing is faster in younger than older mice as in humans.

– The quality of the gross photos in Figure 2E is poor. Therefore, it is highly recommended to replace them with better photos.

– Figures 2F and 2G show something that it is concerning. The photos intend to show the higher intensity of the AF647-ZOL-DNV signal after topical application. First, the photos of the controls show moderate red fluorescent background. Second, the photos for the different doses of AF647-ZOL-DNV show higher intensity but primarily in the calcified tissues of the teeth rather than in the maxillary bones. The signal intensity observed in maxillae appears to be the resultant irradiation from the molars rather than from the maxillary bone. In addition, the figure legend expresses that the signal is after the topical application. What does immediately after mean? Was it immediately after the application or after 2 wks post-extraction? The latter is an important issue that should be clarified.

– Figure 4B: the authors claim to depict an unhealed open wound. I interpret the images differently, with gingival coverage but over a depressed palatal surface, as expected in a healed wound. In my experience, the ulcerated mucosa looks very different. Were these lesions corroborated/validated with histopathology?

– The quality of MicroCT images in Figure 4E is poor. In addition, what are the white arrows depicting?

– What is the region of interest to measure the osteonecrosis depicted in Figure 4F? A description of the analyzed area should be included in the method. Furthermore, histologic photomicrographs supporting these findings would be a good addition.

– According to the study, HMDP-DNV has a high affinity for the hydroxyapatite but does not inhibit the FPPS and does not interfere with the mevalonate pathway and protein prenylation, as the active N-BPs like ZA. Though this is the mechanism to explain the rescue of bone resorption, what would be the mechanism that explains the effects of HMDP-DNV on the immune cells as presented in the single cell RNA sequencing data in Figures 5 and 6?

– Following up on the previous point, how to conceal that systemic treatment with N-BPs (e.g., ZA) has systemic effects on the immune system, but the HMDP-DNV treatment can locally modify the immune cell phenotype? Adaptive immune functions and inflammation are not just "local" events but involve systemic events, where lymphocytes, monocytes, neutrophyls, etc are produced and activated at different organs, recirculate, and are homed at specific organs and tissues.

– In addition, the authors found that N-BP chemisorption to HAp was not permanent but rather a dynamic molecular equilibrium on the bone surface. However, It is unclear how a compound like HMDP-DNV could affect the inflammatory cells in the oral wound.

– BRONJ and DRONJ are clinically and histopathologically indistinguishable and share the same antiresorptive mechanism. Is the anti-RANKL antibody expected to induce the same effects to immune cell function as N-BPs?

– It is well-established clinically and preclinically that oral infection is a risk factor for BRONJ. However, this study never mentioned infection as part of the BRONJ pathogenesis. Indeed, most of the rodent models of BRONJ or DRONJ that involve tooth extraction without associated infection fail to develop the signature of the disease, which is necrotic exposed bone.

– Even though the dose of HMDP for Tc-99m HMDP administration for PET scan is 0.5 mg/mL but also considering that HMDP-DNV competitively displaces ZA from the hydroxyapatite in calcified tissues, would it be concerning for patients undergoing cancer and being treated with ZA to use 99mTc-HMDP PET/CT for diagnosis and follow-ups?

– In results, page 15, lines 296-298, the authors wrote: "Histological evaluations of mouse tooth extraction wound showed delayed bone healing and epithelial hyperplasia leading to the exposure of jawbone in ZOL-treated mice consistent with BRONJ lesion in humans. However, the description of these findings is confusing since wound healing involves epithelial hyperplasia and wound coverage. Besides, Figure 5A does not show any difference in the oral/gingival epithelium, and the hyperplasia is not evident in the photo.

– In the IL-17F and FOXP3 graphs of Figure 5E, the symbols for the No treatment and HMDP-DNV mice groups cannot be distinguished. It would be helpful to modify these graphs to make the data more noticeable.

– The authors postulate that the lack of viable osteoclast as a local immune coordinator may cause the dysregulation of the oral barrier immunity, leading to BRONJ and DRONJ. Moreover, the jawbone N-BP appears to reduce oral osteoclast adversely. However, the study does not show any osteoclast quantification. Furthermore, this is the opposite of most clinical and preclinical studies showing that osteoclast number does not change or even increase in individuals treated with bisphosphonates…

There are a few problems with this. First, it is known that the treatment does not reduce the number of osteoclasts. Indeed, clinical and preclinical studies have shown that, in contrast, osteoclast number does not change or is even increased, though the cells are sometimes detached from the bone surfaces.

– There is no information in the Methods about the number of mice per group in the BRONJ modeling experiments.

---

## [Author Response]

Reviewer #1 (Recommendations for the authors):A minor weakness is that BRONJ can occur clinically in the setting of suppressed inflammation with the use of glucocorticoids- arguing against a purely pro-inflammatory mechanism.Given that BRONJ usually happens in the setting of high frequency doses of BPs, the authors could assess whether HMDP-DNV displacement works in that setting too, which is probably the most clinically relevant one.

As pointed out, glucocorticoids are widely used as combination therapies for multiple myeloma and breast cancer with the use of N-BP and other chemotherapy. Since their discovery in 1940s, glucocorticoids are prescribed as anti-inflammatory agent and the mechanism of action is mediated through the glucocorticoid receptor. Because the glucocorticoid receptors are expressed by many different cells, there is considerable heterogeneity in therapeutic effects. Although glucocorticoids exhibit anti-inflammatory effect, studies reported that glucocorticoid treatment could exacerbate the peripheral immune response and increase the pro-inflammatory cytokine expression. Furthermore, glucocorticoids may directly interact with osteocytes and induce apoptosis causing nontraumatic osteonecrosis.

While the pathological role of glucocorticoids has not been fully established, concomitant glucocorticoid medication with N-BP has been postulated to contribute to the severity of BRONJ, possibly as an immunosuppression agent. Although it is beyond the current scope of our investigation, it will be important to address whether HMDP-DNV displacement would work in the clinically relevant polypharmacy condition. The Discussion section has been revised.

Finally, it would be interesting to note if there is evidence for pro-inflammatory signaling at other mucosal sites in the mice with BP treatment.

This is an important comment. We examined the gingival histology on the non-tooth extraction side. It was clearly observed that without the oral insult created by tooth extraction, gingival inflammation was not observed. The revised manuscript includes new Figure 5 reporting the unwounded jawbone.

Besides oral mucosa, it is well documented that N-BP has the gastrointestinal adverse effects such as erosions and ulcers in stomach and small intestine. The mechanism of gastrointestinal irritation is not well understood. However, N-BP taken by oral administration is though to directly interact with the phospholipids lining the luminal side of gastrointestinal tract. Therefore, the mechanism appears to be distinct from BRONJ. This section was beyond the scope of the current study and was not included in the revision.

Reviewer #2 (Recommendations for the authors):Specific concerns and suggestions for improvement are given below:– A major weakness of this study is that it only investigates the effects of treatments 2 wks post-extraction. The ONJ-related exposed bone is a chronic manifestation of antiresorptive medication. In clinical studies, as the authors mentioned, the efficacy of treatments to treat BRONJ related oral wounds takes a long time, even years.– The authors justified the evaluation of the HMDP-DNV treatment for only 2 wks based on the experience that providing a soft diet post-extraction contributes to the oral wound's healing process; however, as it is well-established, the uneventful wound healing (without infection) of the extraction socket in a rodent could take several weeks and even months to complete if bone healing is taken into account. Moreover, if we consider that N-BPs delay healing, an experiment that only investigates the model for two weeks is insufficient; therefore, it is likely that the model is investigating the effects on early healing but not BRONJ.

The original manuscript presented the abnormal tooth extraction wound healing in mice with ZOL treatment at 2 weeks and 4 weeks after extraction. We performed an additional experiment, in which the outcome of HMDP-DNV topical treatment to the tooth extraction wound of ZOL-injected mice was obtained after 4 weeks of the tooth extraction. In addition to the reported effect of HMDP-DNV at 2 weeks after tooth extraction, this additional experiment addressed (1) the longer study period and (2) more than one time point for treatment assessment.

The complete tooth extraction socket wound healing was reported to occur in 21 days (3 weeks) in mice and 12-24 weeks in humans. Therefore, the additional experiment of 4 weeks in mice should be equivalent to 16-32 weeks (3-8 months) in humans. The new experiment revealed that the HMDP-DNV treatment normalized the tooth extraction wound healing of ZOL-treated mice at 2 weeks and 4 weeks of tooth extraction, whereas ZOL-treated mice without HMDP-DNV topical treatment sustained the abnormal tooth extraction wound healing. As such, new long-term data ensured the lasting treatment effect of HMDP-DNV.

– Another limitation of the study is that it only tested a one-time point. If the treatment effect of HMDP-DNV, as the authors implied, were not to revitalize the necrotic bone but to remove the affected bone by normalizing the osteoclastic activity and halting further necrosis, earlier time points would be needed to substantiate these statements. In addition, to investigate BRONJ development and progression later time points would have been important.

We value this comment. In fact, the new data at 4 weeks after tooth extraction demonstrated a substantial bone removal from the affected alveolar bone, resulting in the decreased osteonecrosis area.

– The authors stated that BRONJ is predominantly induced by the oral N-BP and suggested that HMDP-DNV is a possible effective therapy for BRONJ. Then, what would be the explanation that only 0.005 % and 1-5% of the patients taking osteoporosis or oncologic doses of N-BPs get BRONJ but not all of them?

The present study examined the pathological mechanism of BRONJ. As pointed out, not all patients exposed to N-BP develop BRONJ. In fact, the incidence of BRONJ is quite small. Therefore, other concurrent factors may be involved to cause BRONJ. In our mouse model, BRONJ-like lesion was observed only at the tooth extraction area and the contralateral unwounded side did not show any noticeable abnormality, suggesting that the dentoalveolar surgery played an important role in the mouse BRONJ lesion. A recent multicenter retrospective survey reported that dentoalveolar surgery and tooth extraction increased the risk of BRONJ.

N-BP adsorbed on bone mineral stays inactive until bone resorption by osteoclasts, which release N-BP from bone surface to the local environment. Tooth extraction induces not only bone remodeling in the bony socket but also unusual osteoclastogenesis on the external surface of alveolar bone. It is highly conceivable that osteoclasts on the alveolar bone surface could release N-BP to the oral environment. The present study demonstrated the oral barrier inflammation of ZOL-treated mice, which was densely localized adjacent to the alveolar bone surface. It has been reported that N-BP mediates M1 macrophage polarization and impairs neutrophil function. The single cell RNA sequencing analysis in the present study suggested the gene expression signature of proinflammatory myeloid cells, which was reversed to anti-inflammatory phenotypes by the HMDP-DNV treatment removing ZOL, suggesting that prolonged oral inflammation in the BRONJ lesion might be induced by released N-BP in the oral barrier tissue.

Separately, the present study demonstrated that osteoclasts on the alveolar bone surface in ZOL-treated mice were found to form abnormally shallow bone resorption lacunae, suggesting that osteoclasts internalized N-BP were likely to be inactivated. The HMDP-DNV treatment appeared to normalize the osteoclastic activity generating large resorption lacunae, resulting in the removal of necrotic bone at 4 weeks after tooth extraction. The resolution of BRONJ often occurs after the necrotic bone is sequestrated. It is tempting to speculate that bone sequestration might have occurred in HMDP-DNV treated mice between 2 weeks and 4 weeks of tooth extraction.

It must be noted that single treatment of HMDP-DNV did not prevent the development of BRONJ. Therefore, we postulate that there is a threshold of the bioavailable N-BP dose on the jawbone required to cause BRONJ. The repeated HMDP-DNV treatments may have decreased the ZOL dose levels below the threshold to induce BRONJ even with dentoalveolar surgery.

In conclusion, the present study suggests that the above threshold dose of N-BP adsorbed on the jawbone may constitute one of the critical factors. However, the pathological mechanism of BRONJ may involve multiple events, which may explain why not all patients with the history of N-BP treatment develop BRONJ. The Discussion section was revised.

– Since bisphosphonates' main effects inhibit osteoclast and bone resorption, it is surprising that osteoclasts and bone resorption were not quantitatively assessed in the alveolar bone tissues in the in vivo study.

We performed an additional experiment. To determine bone resorption endpoints, we performed Immunohistochemical experiments using anti-Cathepsin K (Ctsk) antibody to clearly identify osteoclasts. We found that untreated jawbone of ZOL-treated mice showed deformed Ctsk-positive osteoclasts that were persistently observed until 4 weeks of tooth extraction healing. By contrast, the HMDP-DNV-treated jawbone of ZOL-treated mice showed normalized Ctsk-positive osteoclasts at the early wound healing period (2 weeks). At the long-term wound healing period (4 weeks), osteoclasts were not observed. New Figure 5 was created.

– In the introduction, the authors presented an alternative MRONJ "prevalence" estimation of approximately 36,000 new cases per year (Table 1). They claimed that still considered within the FDA's orphan disease definition of under 200,000 annually. There are two possible issues with this sentence. First, the definition states 200,000 people, BUT the estimation is not per year but at a specific time. Second, it is incorrect to refer to prevalence. Prevalence is the proportion of people in a population with a particular disease or condition. Therefore, it is more appropriate to call it incidence since it refers to "new cases" of disease in a population over a specific period.

We revised the “prevalence” to “incidence.” The Orphan Drug Act of 1983 defines a rare disease as a disease or condition that affects less than 200,000 people in the US. Considering a long cure period, the cumulative number of MRONJ patients may exceed this annual estimation; however, it is highly conceivable that MRONJ is a rare disease defined by the FDA’s rare disease. The introduction section was revised.

– In the introduction, the authors claimed that HMDP-DNV topical applications to "nascent mouse BRONJ lesions" resulted in accelerated gingival wound closure and bone socket healing as well as attenuation of osteonecrosis development. This statement is inaccurate because the BRONJ lesion never developed in the HMDP-DNV treated group, according to the data presented at 2 wks. Therefore, it appears to be more a "preventive approach." In order to make this statement, earlier time points would be required to substantiate the "nascent BRONJ lesions" developed in HMDP-DNV.

As we demonstrated in Figure 4, prior to HMDP-DNV topical application, after the tooth extraction, the mice exhibited abnormal healing as observed as delayed soft tissue closure and osteonecrosis. Therefore, in our study design, HMDP-DNV was applied to nascent BRONJ lesion or early signs of BRONJ.

– Please add a reference for the FRAX score (line 62-64).

The FRAX fracture risk assessment tool was developed by the World Health Organization. We added Kanis et al. (2007) Osteoporos Int, 18:1033-1046. (PMID: 17323110)

– Refs 5 and 9 are only for osteoporosis doses (lines 66-67) but not the oncologic dose in the abstract.

The references were replaced by Gordon (2005) Clin Breast Cancer 6(2):125-31 (PMID 16001990)

– Considering that BRONJ affects mature adults, why did the authors utilize 8-10 wks old mice? The wound healing is faster in younger than older mice as in humans.

Tooth extraction from mice presented a challenging model development. It was important to prevent the traumatic tooth extraction, which caused a serious confounding factor for sound tooth extraction wound healing. The younger mice at 8-10 wks allowed consistent atraumatic tooth extraction in our study. The methods section was revised.

– The quality of the gross photos in Figure 2E is poor. Therefore, it is highly recommended to replace them with better photos.

Figure 2E has been replaced.

– Figures 2F and 2G show something that it is concerning. The photos intend to show the higher intensity of the AF647-ZOL-DNV signal after topical application. First, the photos of the controls show moderate red fluorescent background. Second, the photos for the different doses of AF647-ZOL-DNV show higher intensity but primarily in the calcified tissues of the teeth rather than in the maxillary bones. The signal intensity observed in maxillae appears to be the resultant irradiation from the molars rather than from the maxillary bone. In addition, the figure legend expresses that the signal is after the topical application. What does immediately after mean? Was it immediately after the application or after 2 wks post-extraction? The latter is an important issue that should be clarified.

While AF647-ZOL-DNV assisted the trans-oral epithelial delivery, some AF647-ZOL-DNV appeared to be disintegrated in the oral cavity. The released AF647-ZOL might be adsorbed to the exposed tooth surface or the enamel structure. Thus, we set the ROI only on the palatal bone without the tooth structure. The methods section was revised.

– Figure 4B: the authors claim to depict an unhealed open wound. I interpret the images differently, with gingival coverage but over a depressed palatal surface, as expected in a healed wound. In my experience, the ulcerated mucosa looks very different. Were these lesions corroborated/validated with histopathology?

We appreciate the suggestion. In the revised manuscript, the corresponding histology was added as a new Figure 5, which clearly demonstrated the open gingival wound.

– The quality of MicroCT images in Figure 4E is poor. In addition, what are the white arrows depicting?

MicroCT image of Figure 4E was replaced. The white arrows indicated that unhealed extraction socket without generating bone.

– What is the region of interest to measure the osteonecrosis depicted in Figure 4F? A description of the analyzed area should be included in the method. Furthermore, histologic photomicrographs supporting these findings would be a good addition.

The new Figure 5 exhibits the histological data.

– According to the study, HMDP-DNV has a high affinity for the hydroxyapatite but does not inhibit the FPPS and does not interfere with the mevalonate pathway and protein prenylation, as the active N-BPs like ZA. Though this is the mechanism to explain the rescue of bone resorption, what would be the mechanism that explains the effects of HMDP-DNV on the immune cells as presented in the single cell RNA sequencing data in Figures 5 and 6?– In addition, the authors found that N-BP chemisorption to HAp was not permanent but rather a dynamic molecular equilibrium on the bone surface. However, It is unclear how a compound like HMDP-DNV could affect the inflammatory cells in the oral wound.– Following up on the previous point, how to conceal that systemic treatment with N-BPs (e.g., ZA) has systemic effects on the immune system, but the HMDP-DNV treatment can locally modify the immune cell phenotype? Adaptive immune functions and inflammation are not just "local" events but involve systemic events, where lymphocytes, monocytes, neutrophyls, etc are produced and activated at different organs, recirculate, and are homed at specific organs and tissues.

As stated in critique [4], the present study demonstrated the oral barrier inflammation of ZOL-treated mice, which was localized adjacent to the abnormal osteoclasts. It is highly conceivable that osteoclasts could released N-BP to the oral environment. It has been reported that N-BP mediates M1 macrophage polarization and impairs neutrophil function. The single cell RNA sequencing analysis in the present study suggested the gene expression signature of proinflammatory myeloid cells, which was reversed to anti-inflammatory phenotypes by the HMDP-DNV treatment, suggesting that prolonged oral inflammation in the BRONJ lesion might be induced by released N-BP in the oral barrier tissue.

Recently, the presence of osteoclasts was needed to resolve oral barrier inflammation. We have shown that osteoclasts secrete anti-inflammatory cytokines and the exposure to N-BP modified to increase the secretion of pro-inflammatory cytokines. Therefore, the presence of N-BP-affected osteoclasts may have contributed to oral barrier inflammation.

It has been established that N-BP affects systemic inflammation. Immediately after IV infusion, N-BP is internalized in circulating macrophages, which release the isopentenyl pyrophosphate (IPP), an intermediate molecule of the interfered mevalonate pathway. IPP is a robust ligand of circulating human gd T cells and the activated gd T cells cause a transient flu-like syndrome. We have previously reported that the alteration of spleenocytes, bone marrow immune cells and circulating immune cells in ZOL-treated mice. However, in these mice, BRONJ was not developed until tooth extraction wounding.

The Discussion section was revised.

– BRONJ and DRONJ are clinically and histopathologically indistinguishable and share the same antiresorptive mechanism. Is the anti-RANKL antibody expected to induce the same effects to immune cell function as N-BPs?

This is beyond the scope of the current investigation. However, we postulate that the removal of healthy osteoclastic functions by denosumab and N-BP may underlie the common pathological mechanism. The Discussion section was revised.

– It is well-established clinically and preclinically that oral infection is a risk factor for BRONJ. However, this study never mentioned infection as part of the BRONJ pathogenesis. Indeed, most of the rodent models of BRONJ or DRONJ that involve tooth extraction without associated infection fail to develop the signature of the disease, which is necrotic exposed bone.

AAOMS recommend antibiotics and chlorhexidine rinses for patients with Stage 2 MRONJ with a sign of infection. The therapeutic containment of infection was shown to decrease pain of the MRONJ lesion. However, the effect of antibiotics on the resolution of MRONJ has not been fully established. Therefore, it is still unclear if oral infection is a causal factor of BRONJ.

– Even though the dose of HMDP for Tc-99m HMDP administration for PET scan is 0.5 mg/mL but also considering that HMDP-DNV competitively displaces ZA from the hydroxyapatite in calcified tissues, would it be concerning for patients undergoing cancer and being treated with ZA to use 99mTc-HMDP PET/CT for diagnosis and follow-ups?

In a clinical report, bone scintigram using Tc-99m HMDP was better identified MRONJ lesion than Tc-99m MDP. In bone scintigram diagnosis, Tc-99m HMDP is IV injected. Therefore, the local dose of the jawbone is unclear. We developed a topical intraoral formulation ensuring the local oral dose of HMDP. Because the topical HMDP-DNV topical application is non-invasive treatment, we envision the repeated applications in a translational and clinical investigation. The Discussion section was revised.

– In results, page 15, lines 296-298, the authors wrote: "Histological evaluations of mouse tooth extraction wound showed delayed bone healing and epithelial hyperplasia leading to the exposure of jawbone in ZOL-treated mice consistent with BRONJ lesion in humans. However, the description of these findings is confusing since wound healing involves epithelial hyperplasia and wound coverage. Besides, Figure 5A does not show any difference in the oral/gingival epithelium, and the hyperplasia is not evident in the photo.

The revised manuscript contains more photomicrographs to highlight our observation in Figure 5.

– In the IL-17F and FOXP3 graphs of Figure 5E, the symbols for the No treatment and HMDP-DNV mice groups cannot be distinguished. It would be helpful to modify these graphs to make the data more noticeable.

The figure was revised as suggested.

– The authors postulate that the lack of viable osteoclast as a local immune coordinator may cause the dysregulation of the oral barrier immunity, leading to BRONJ and DRONJ. Moreover, the jawbone N-BP appears to reduce oral osteoclast adversely. However, the study does not show any osteoclast quantification. Furthermore, this is the opposite of most clinical and preclinical studies showing that osteoclast number does not change or even increase in individuals treated with bisphosphonates…There are a few problems with this. First, it is known that the treatment does not reduce the number of osteoclasts. Indeed, clinical and preclinical studies have shown that, in contrast, osteoclast number does not change or is even increased, though the cells are sometimes detached from the bone surfaces.

The revised manuscript included the quantitative evaluation of osteoclasts. The HMDP-DNV treatment decreased a total number of osteoclasts on the jawbone surface, in particular, from the distant area from the tooth extraction socket.

– There is no information in the Methods about the number of mice per group in the BRONJ modeling experiments.

The revision was made as suggested.